# Identification of long regulatory elements in the genome of *Plasmodium falciparum* and other eukaryotes

**Christophe Menichelli**[1], **Vincent Guitard**[2], **Rafael M. Martins**[2], **Sophie Lèbre**[3,4], **Jose-Juan Lopez-Rubio**[2]*, **Charles-Henri Lecellier**[1,5]*, **Laurent Bréhélin**[1]*

**1** LIRMM, Univ Montpellier, CNRS, Montpellier, France, **2** Laboratory of Pathogen-Host Interactions (LPHI), UMR5235, CNRS, Montpellier University, INSERM, Montpellier, France, **3** IMAG, Univ. Montpellier, CNRS, Montpellier, France, **4** Univ. Paul-Valéry-Montpellier 3, Montpellier, France, **5** Institut de Génétique Moléculaire de Montpellier, University of Montpellier, CNRS, Montpellier, France

\* jose-juan.lopez-rubio@inserm.fr (JJLR); charles.lecellier@igmm.cnrs.fr (CHL); brehelin@lirmm.fr (LB)

**Data Availability Statement:** The source code (python) of DExTER is available at address https://gite.lirmm.fr/menichelli/DExTER. This git repository

## Abstract

Long regulatory elements (LREs), such as CpG islands, polydA:dT tracts or AU-rich elements, are thought to play key roles in gene regulation but, as opposed to conventional binding sites of transcription factors, few methods have been proposed to formally and automatically characterize them. We present here a computational approach named DExTER (Domain Exploration To Explain gene Regulation) dedicated to the identification of candidate LREs (cLREs) and apply it to the analysis of the genomes of *P. falciparum* and other eukaryotes. Our analyses show that all tested genomes contain several cLREs that are somewhat conserved along evolution, and that gene expression can be predicted with surprising accuracy on the basis of these long regions only. Regulation by cLREs exhibits very different behaviours depending on species and conditions. In *P. falciparum* and other Apicomplexan organisms as well as in *Dictyostelium discoideum*, the process appears highly dynamic, with different cLREs involved at different phases of the life cycle. For multicellular organisms, the same cLREs are involved in all tissues, but a dynamic behavior is observed along embryonic development stages. In *P. falciparum*, whose genome is known to be strongly depleted of transcription factors, cLREs are predictive of expression with an accuracy above 70%, and our analyses show that they are associated with both transcriptional and post-transcriptional regulation signals. Moreover, we assessed the biological relevance of one LRE discovered by DExTER in *P. falciparum* using an *in vivo* reporter assay.

The source code (python) of DExTER is available at https://gite.lirmm.fr/menichelli/DExTER.

## Author summary

Gene expression is regulated at different levels and by different mechanisms in Eukaryotes. At the DNA level, transcription factors (TFs) are supposed to play a key role by binding short motifs in promoters or enhancers. In *Plasmodium falciparum*, the causative

also provides the R scripts for reproducing the main experiments described in the paper.

**Funding:** The work was supported by funding from CNRS (International Associated Laboratory "miREGEN", C-H.L. & L.B.), INSERM-ITMO Cancer (BIO2015-04 "LIONS", C-H.L. & S.L. & L.B.), Plan d'Investissement d'Avenir (#ANR-11-BINF-0002 "Institut de Biologie Computationnelle", C-H.L. & S. L. & L.B. and #ANR-11-LABX-0024-01 "ParaFrap", J-J.L-R. & V.G.), Labex NUMEV (GEM Flagship project, C-H.L. & S.L. & L.B.), CNRS/INSERM funding Défi Santé numérique (project REGAI, C-H. L.), the Fondation pour la Recherche Médicale (DEQ2018033199, J-J.L-R. & R.M.M.), and the program ATIP-Avenir (J-J. L-R.) The funders had no role in study design, data collection and analysis, decision to publish, or preparation of the manuscript.

**Competing interests:** The authors have declared that no competing interests exist.

agent of severe malaria in humans, different levels of gene regulation are also present, but very few TFs have been identified and validated so far. We propose here a computational method for the identification of a new type of regulatory elements called long regulatory elements (LRE). Contrary to TF motifs, that are usually 6-12bp long, LREs may span dozens or hundreds of base pairs. Moreover, no computational method have been specifically dedicated to their identification until now. We show with our method that, depending on species and conditions, LREs may play important role in gene regulation. For *P. falciparum*, these elements appear to determine a very large part of gene expression variation in all stages of the parasite life cycle.

## Introduction

Gene expression is regulated at different levels and by different mechanisms in Eukaryotes. At the DNA level, transcription factors (TFs) are supposed to play a key role by binding to specific motifs of typically 6-12 bp in promoters or enhancers. However, TFs are not the only actors, and other mechanisms such as histone occupation, epigenetic marks, transcript stability, 3D structure of the chromatin, etc. are known to be involved in the whole and entangled process of gene expression regulation.

In *Plasmodium falciparum*, the causative agent of severe malaria in humans, different levels of gene regulation are also present, including cis-regulatory DNA elements, transcription factors, epigenetic regulation, and post- transcriptional and translational control. Recently, around 4000 regulatory elements have been identified by directly profiling chromatin accessibility. The vast majority of these sites are located within 2000 bp upstream of genes and their chromatin accessibility pattern correlates positively with abundance of mRNA transcripts [1]. Main factors of the general transcription machinery are present in the Plasmodium genome, yet only a few specific TFs (mostly belonging to the apicomplexan AP2 TF family) have been identified and validated [2–7]. They constitute approximately 1% of all protein-coding genes [8, 9] compared to ∼ 3% in yeast or 6% in human. Among the mechanisms for epigenetic regulation, covalent histone modifications are the best described so far, and experimental evidences show that this form of regulation is most evident in heterochromatin-mediated silencing of genes located in subtelomeric regions and a few chromosome-internal heterochromatic islands while the largest part of the genome is in an euchromatic transcriptionally permissive state [10–13]. Finally, several studies have shown that post-transcriptional regulation (mRNA degradation) and translational control mechanisms also operate in this parasite (see for example [14–17]).

Study of the links between DNA and gene expression has a long history in bioinformatics. Notably, numerous approaches have been proposed to identify TF motifs by searching for motifs shared by sequences associated with a given gene expression profile [18–24]. In recent years, it has been shown that transcription factor binding but also gene expression as well as several related chromatin features such as histone modifications or DNase I–hypersensitive sites can be predicted from DNA sequence only, often with surprisingly high accuracy [25–30]. With the exception of a few approaches (*e.g.* [26, 30]), deep learning, and particularly convolutional neural networks (CNN), are often used for this task [25, 27, 29, 31, 32]. The good predictive performances of these approaches suggest that a large part of the instructions for gene regulation lies at the level of the DNA. However, identifying the exact DNA features captured by CNNs and assessing their respective predictive power remains a difficult task [33]. Interesting methods are being developed to post-analyze and interpret learned CNNs in order

to identify the DNA determinants used for the predictions (see *e.g.* [25, 33, 34]) but, to the best of our knowledge, these attempts are limited to the identification of single nucleotides and motifs.

Besides conventional motifs of TF binding sites (TFBS), which usually display strong positional information and relatively short length (dozens of bp), several studies have highlighted the role that longer regions without clear motif but with biased nucleotide composition can have for regulating gene expression. The most famous example in vertebrates is CpG islands, which are defined as regions larger than 200bp with a high ratio of CpG dinucleotides. CpG islands often correspond to transcription initiation sites [35]. While the exact mechanism by which they acquire their function remains a debated subject, they are now widely considered as important regulatory structures of mammalian genomes [36]. Similarly, other works have shown that large (hundreds of bp) CpG-rich domains directly downstream of TSS and that do not classify as CpG islands increase transcription rates of endogenous genes in human cells [37]. Short tandem repeats are another class of biased regions that could act as regulatory elements. These elements are made of periodic k-mers of 2-6 bp, spanning regions whose total length has been shown to widely impact gene expression and to contribute to expression variation, independently of their genomic location (exon, intron, intergenic) [38]. PolydA:dT tracts have been shown to act as promoter elements favoring transcription by depleting repressive nucleosomes [39], specifically by orientating the displacement of nucleosomes [40]. Other examples are the AU-rich elements, which are 50-150 nt sequences, rich in adenosine and uridine bases. They are located in the 3'-UTRs of many short half-life mRNAs and are believed to regulate mRNA degradation by a mechanism dependent on deadenylation [41]. Recently, high-resolution chromatin conformation capture (Hi-C) experiments have revealed the existence of contiguous genomic regions with high contact frequencies referred to as topologically associated domains (TADs) [42]. It has been shown that TADs actually correspond to different isochores (*i.e.* large regions with homogeneous G+C content) [43], and that genes within the same TAD tend to be coordinately expressed [44], thus highlighting the role nucleotide composition of large regions may have on the regulation of gene expression. In accordance with these observations, we have shown that gene expression in humans can be predicted with surprising accuracy only on the basis of di-nucleotide frequencies computed in predefined gene regions (close promoters, upstream and downstream promoter regions, 5'- and 3'-UTRs, exons, introns) [26]. Importantly, we observed that although CpG content in promoters has high contribution when predicting gene expression, dinucleotides other than CpG are also important and likely contribute to gene regulation [26]. In line with these works, Quante and Birds have previously proposed that domains with specific base compositions might modulate the epigenome through cell-type-specific proteins that recognize frequent, short k-mers [45]. Likewise, Lemaire et al. have found that exon nucleotide composition bias establishes a direct link between genome organization and local regulatory processes, like alternative splicing [46].

However, while numerous computational approaches have been developed to identify motifs, no methods have been designed for characterizing these long regulatory regions associated with specific nucleotide composition. Especially, *in silico* methods are needed to automatically identify the boundaries and nucleotide specificities (nucleotide, dinucleotide or longer k-mers) of the long regulatory elements (LREs) present in sequenced genomes. It is worth noting that segmentation methods that aim to identify sections of homogeneous composition along the genome have been proposed, for example for identifying CpG islands [47–49]. However, with these approaches the segmentation is done on the basis of the sequence alone, without using gene expression to help in the segment definition. As a consequence, the inferred regions are not specifically linked to gene expression regulation, and the approach can miss several regulatory elements.

Here we propose a new method named DExTER (Domain Exploration To Explain gene Regulation) for identifying the precise boundaries and nucleotide specificities of long regions whose nucleotide content is correlated with gene expression (hereafter denoted as candidate Long Regulatory Elements, or cLREs). More precisely, given a set of sequences (one for each gene) and gene expression data in a given condition (treatment, time point or cell type), DExTER identifies pairs of (k-mer,region) for which there is a correlation between gene expression and the frequency of the k-mer in the defined region of each gene. DExTER uses an iterative procedure to explore the space of (k-mer,region) by gradually increasing the size of the k-mers. The identified pairs form a set of predictive variables that are then combined to predict gene expression. The predictor is trained with machine learning algorithms based on penalized likelihood that allows us to select a minimal number of predictive variables and hence to identify the most important regions and k-mers that could act as regulatory elements.

We applied DExTER on 4000bp sequences centered around the gene start (most upstream TSS) or AUG of *P. falciparum* and several other organisms in the Eukaryote tree. Depending on the species, the method identified different large regions (hundred of bps) whose enrichment in certain k-mers was correlated with gene expression. We hypothesized that these long biased sequences could constitute regulatory elements, different from the classical TF binding sites that usually involve only a dozen base pairs. For most tested species, we show that these cLREs are predictive of expression with an accuracy in between 50% and 60%. For *P. falciparum* the accuracy even exceeds 70%, indicating that such elements could have a predominant role in this species. Furthermore, our analysis showed that this regulation is highly dynamic, with different regions and k-mers involved at different stages of *Plasmodium* life cycle. For species outside of the Apicomplexa phylum, the mechanism appears much more static, except for *Dictyostelium discoideum* and in embryonic development of *Drosophila* and *C. elegans*. Further analysis in *Plasmodium* showed a clear dichotomy between identified cLREs, with elements located upstream of the TSS being mainly associated with transcriptional regulation, while downstream elements being mainly involved post-transcriptionally. Finally, in order to validate one of the identified elements in *P. falciparum*, a GFP reporter assay was performed, confirming our *in silico* results, thus demonstrating the importance of this LRE in *P. falciparum* biology.

## Results

### An in silico approach for identifying long regions linked to expression

The DExTER method takes gene expression data and a set of sequences (one for each gene) aligned on a common anchor. In the experiments below, we took the 4000bp sequences centered either around the gene start (*i.e.* most upstream TSS) or around the AUG though other alignment anchors could also be used (gene end, exon-intron boundaries, etc.). In the first step (feature extraction), DExTER identifies pairs of (k-mer,region) for which the frequency of the k-mer in the defined region is correlated with gene expression. Sequences are first segmented in different bins. We used 13 bins in the following experiments. The number of bins impact the precision of the approach but also the computing time of the analysis. DExTER starts with 2-mer (dinucleotides) and, for each 2-mer, identifies the region of consecutive bins for which the 2-mer frequency in the region is mostly correlated with gene expression. A lattice structure is used for this exploration (see Fig 1 and details in section Materials and Methods). Once the best region has been identified for a 2-mer, DExTER attempts to iteratively extend this 2-mer for identifying longer k-mers. For each considered k-mer, a lattice analysis is run to identify the best region associated with this k-mer. At the end of the process, a set of variables corresponding to the frequency of the identified k-mers in the identified regions are returned for

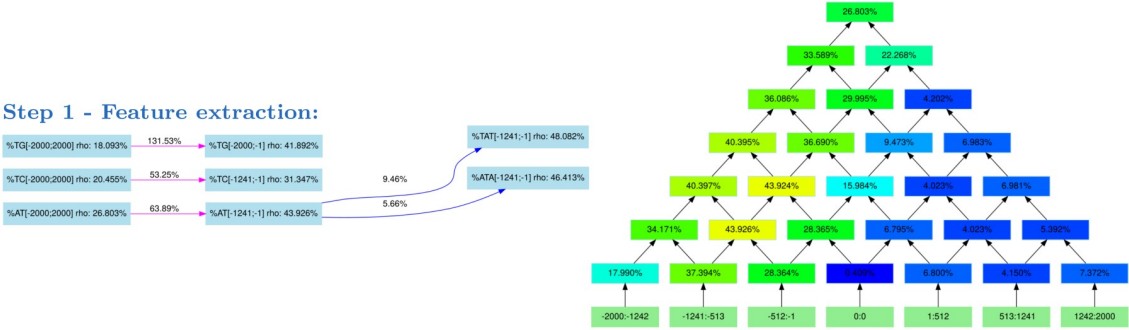

**Fig 1. The DExTER method.** In step 1, DExTER attempts to identify pairs of (k-mer,region) for which the frequency of the k-mer in the defined region is correlated with gene expression. DExTER starts with a 2-mer and compute a lattice (right) representing different regions. The top of the lattice represents the whole sequence, while lower nodes represent smaller regions. At each position, the correlation between 2-mer frequency and gene expression is computed, and regions with highest correlation are identified. For example, in the depicted lattice (which is the lattice associated with k-mer AT) the correlation between gene expression and AT frequency in region [-1241,-1] is 43.926%, while the correlation between expression and the AT frequency in region [513, 1241] is only 4.150%. Then, the 2-mer is extended to 3-mers, and the correlation with expression are computed in the best regions. If the correlation increases, the whole process is repeated with increasing k-mers. Otherwise, DExTER starts a new exploration from a different 2-mer, until every 2-mer has been explored. This way, different variables (*i.e.* pairs of (k-mers-regions)) are iteratively built (see an extract of the exploration graph on the left). In step 2, the frequency of all variables identified in step 1 are gathered into one long table. Then, a linear model predicting gene expression from a linear combination of the variables is learned. A special penalty function (LASSO) is used during training, for selecting only the best variables in the model (blue columns). If several gene expression data are available for one species (*i.e.* several *y* vectors), then step 1 is ran independently on each data, and all identified variables are gathered into a single table. Then, a linear model is learned for each data, but the different models are learned simultaneously with another penalty function that tends to select the same variables for the different data (group LASSO for multitask learning, see Materials and methods).

each gene. In the second step, DExTER learns a model that predicts gene expression on the basis of these variables. We used a linear regression model:

$$y(g) = a + \sum_i b_i x_{i,g} + e(g), \tag{1}$$

where $y(g)$ is the expression of gene $g$, $x_{i,g}$ is variable $i$ for gene $g$, $e(g)$ is the residual error associated with gene $g$, $a$ is the intercept and $b_i$ is the regression coefficient associated with variable $i$. Because the set of variables identified in the first step may be large and variables are often correlated, the model is trained with a lasso penalty function [50] that selects the most relevant variables solely (feature selection). Finally, once a model has been trained, its accuracy is evaluated by computing the correlation between predicted and observed expressions on several hundred genes. To avoid any bias, this is done on a set of genes that have not been used in the two previous steps.

## Long sequences with specific composition are predictive of expression for several Eukaryotes and especially for *P. falciparum*

The approach has been applied to several series of expression data, targeting unicellular and multicellular eukaryotes in different conditions. Besides the erythrocytic cycle of *P. falciparum* [51], we also studied the *P. berghei* life cycle [52] as well as that of *T. gondii* [53], another species in the Apicomplexan taxa to which belong the two *Plasmodium*s. The *S. cerevisiae* response to NaCl stress [54] completes the comparisons for unicellular organisms. For

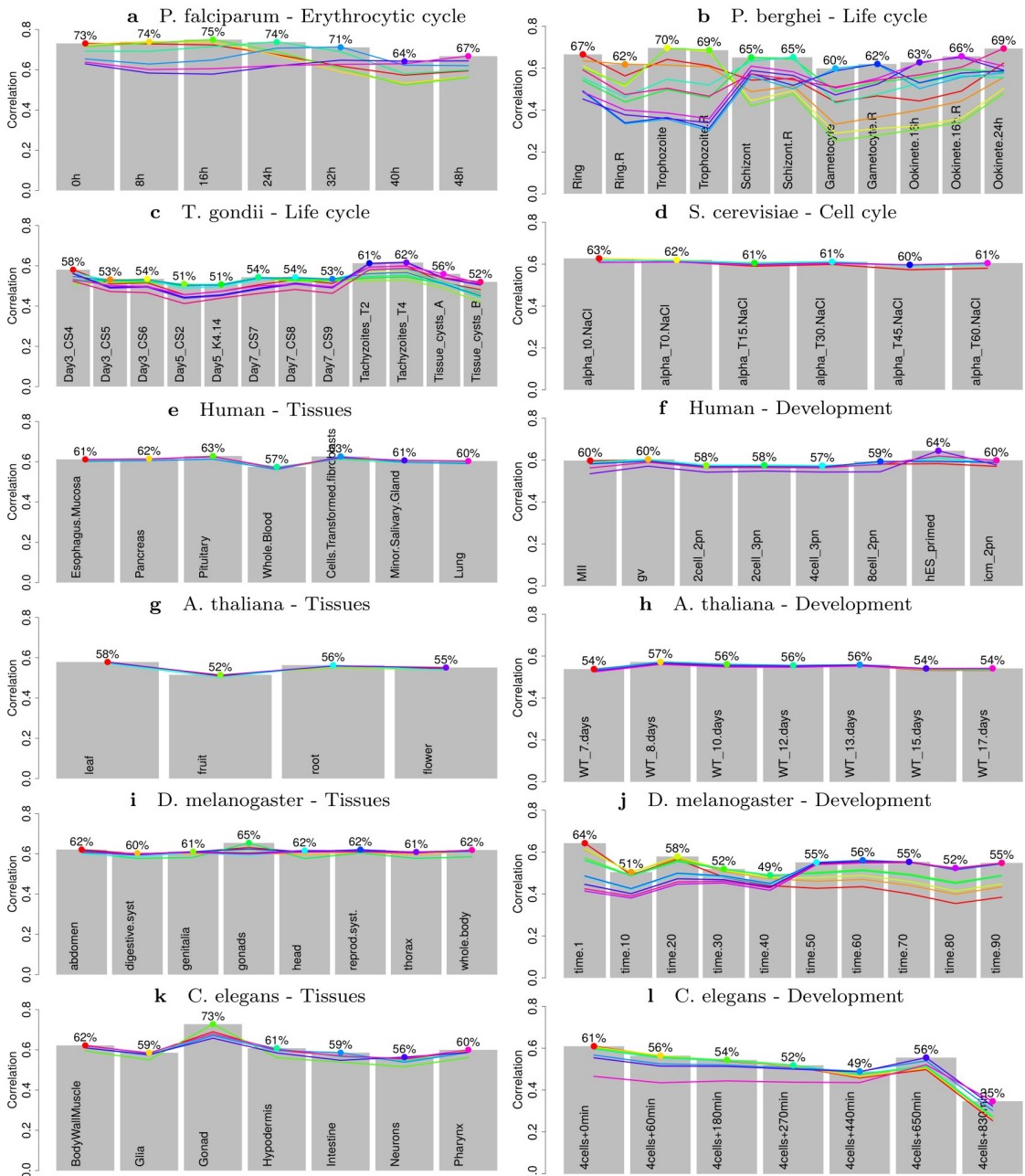

**Fig 2. Accuracy of the DExTER models for predicting coding-gene expression in different species and conditions.** Grey charts represent the accuracy, measured as the correlation between predicted and observed gene expression, of the models learned on different conditions. Colored curves summarize the accuracy of a model learned on a specific condition (identified by a big dot of the same color) when used to predict the other conditions of the same series.

multicellular organisms, two series monitoring tissues and development were analyzed for *Drosophila* [55, 56], *C. elegans* [57, 58], human [59, 60], and the plant *A. thaliana* [61, 62]. Note that one model is learned for each condition, hence several models are learned for each series using multitask learning (see Materials and methods). In all experiments, 2/3 genes are used for training (steps 1 and 2) and 1/3 genes are used for measuring accuracy (the same training/testing gene sets are used in all conditions of the same series). Bar charts on Fig 2

summarize the accuracy (Pearson correlation between predicted and measured gene expression) achieved in the different conditions. Except for *P. falciparum* and *P. berghei*, all sequences are centered around the gene start (*i.e.* most upstream TSS). For *Plasmodium* species, we obtained higher accuracies with sequences centered on the AUG, so we used this anchor in the experiments below. Note that this is in concordance with the results of Read et al. [63], who also obtained better accuracy using AUGs rather than TSSs when predicting *P. falciparum* gene expression from epigenetic marks.

The accuracy of the method fluctuates around 60% for most species, thus generalizing our previous study on human that showed that gene expression can be predicted with surprisingly high accuracy using only nucleotide frequencies of specific large gene regions [26]. Intriguingly, for *P. falciparum* the accuracy exceeds 70% on many stages, which is particularly interesting in an organism for which most attempts for identifying TFs have been unsuccessful.

For the sake of comparison, we also learned series of models using only the frequencies of the 16 dinucleotides computed on the whole sequence as predictive variables. The results are reported on S1 Fig. As we can see, the accuracy drops up to 40% in certain cases, highlighting the importance of higher level k-mers and of the proper definition of region boundaries for predicting gene expression. Next, we also learned series of models that use TF binding motif scores instead of DExTER variables (see Materials and methods), and another series that use both TF motif scores and DExTER variables (S2 Fig). For these experiments, we used the 23 ApiAP2 motifs identified in Campbell et al. [64] for *P. falciparum*, and the JASPAR 2020 [65] Fungi, Vertebrates, Plants, and Insects motifs databases for *S. cerevisiae*, human, *A. thaliana* and *Drosophila*, respectively. Here again, we observe a substantial drop when using only motif scores as predictive variables (above 20% for most species and conditions, except for human). On the other hand, adding motif scores to DExTER variables only marginally increases the accuracy. These results highlight the importance of long regions in comparison to (short) motifs for predicting gene expression. In addition, it is important to note that by combining the scores of several motifs, the motif-only models can actually also capture k-mer frequencies —for example, a GC-rich sequence will get high scores for GC-rich motifs. Hence, part of the signal captured by these models is likely related to k-mer frequencies as well.

We then asked whether these long regions detected by our approach may correspond to multiple occurrences of classical TF binding motifs. To do so, we concentrated on the 5 most important variables identified by the learning procedure in each condition (see Materials and methods for details). Fig 3 reports the distribution of the lengths of k-mers and regions of

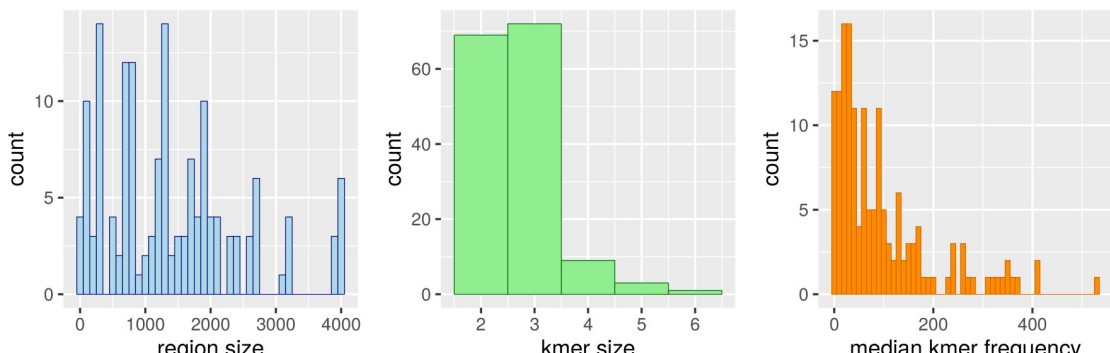

**Fig 3. Lengths and frequencies of the variables identified in the different species and conditions.** The left histogram reports the distribution of k-mer lengths of the most important variables identified in all species and conditions, while the middle histogram reports the distribution of region lengths of these variables. The right histogram reports the median number of occurrences of the identified k-mers in the identified regions in all studied sequences of the different species.

these variables, as well as the distribution of the median number of k-mer occurrences in the identified region of the sequence. In most cases, the large size of the regions (hundreds base pairs), the shortness of the k-mers (3 or less) and the high number of occurrences (median number > 20) seem incompatible with classical TFBSs, which usually involve a dozen base pairs and, to the best of our knowledge, are not known to repeat such high number of times on such long regions. Actually, from the 154 studied variables, we estimate that less than a dozen may correspond to traditional TFBS motifs. A notable exception is the k-mer AGACA identified in *P. berghei*, and whose frequency in the identified region is either 1 or 0 for most sequences. Apart from other variants of this k-mer (which represent most of the identified exceptions) one can also distinguish the short motif TTA, whose presence at the exact position of gene TSSs in *C. elegans* seems negatively correlated with expression.

We then studied more closely the way k-mer occurrences are distributed along the identified regions. We checked whether the occurrences tend to appear isolated or inside repeat blocks. We analyzed for this the most important variable of each species. S3 Fig reports for each species and each gene the proportion of occurrences that appear isolated, while S4 Fig reports the histogram of repeat block length. For these analyses, a repeat block involves at least two k-mer occurrences that are either immediately consecutive or overlapping (for example, sequences ATAATA and ATATA are two blocks made up of two repeats of the ATA k-mer). Except for the two represented *Plasmodium* variables, more than 75% of k-mer occurrences are isolated, and the few repeat blocks are mainly made up of two repeats. The picture is different for the two variables in *Plasmodium* species, as isolated occurrences seem much rarer (around 25%), and the size of a typical repeat block usually fluctuates in between 2 and 20 repeats.

## Dynamics, composition, and location of cLREs differ depending on species and conditions

We next sought to test whether cLREs are associated with dynamic or static regulation processes. For this, each model learned in a specific condition was used to predict expression in other conditions of the same series, and accuracy was measured. Colored curves on Fig 2 summarize these permutation experiments. Static and dynamic behaviors appear to coexist and to be highly dependent on species and conditions. While approximately the same model is learned on the different tissues of human, *A. thaliana*, *Drosophila* and *C. elegans*, in the two *Plasmodium* species, a model learned on a specific stage has poor accuracy on the other stages, suggesting that it is not the same regulatory elements that are used at these different stages. For *T. gondii* the behavior is similar, although much less strong, while for *S. cerevisiae* the mechanism seems completely static. Interestingly, a dynamic behavior is also observed on developmental series of *C. elegans* and *Drosophila* although almost no differences can be observed when permuting the models learned on different tissues of these organisms. Note however that, for both species, slight differences can be observed in gonads, and that this tissue is also the one where the accuracy is the highest. Part of the differences among species can be explained by the inherent correlation of expression between conditions, which are, on average, higher in human, *Drosophila*, *C. elegans*, *A. thaliana* and yeast than in *Plasmodium* and *T. gondii* (see S5 Fig). However, this does not seem to be the only reason for the static behavior observed in former species. Indeed, when we restrict the comparisons to pairs of conditions with similar expression correlations, *Plasmodium* models are still more different than tissue models of human, *Drosophila* or *C. elegans*. For example, the 0h/48h *P. falciparum* pair, and the whole-blood/pituitary human pair have both expression correlation around 80% but, contrary to *P. falciparum*, the human models seem completely interchangeable. Similarly, several

pairs of tissues of *A. thaliana* and *C. elegans* show only moderate expression correlations, although approximately the same model is learned on these tissues.

We next sought to compare the composition and location of regions identified in the different species and conditions. For this, we concentrated on the 5 most important variables identified by the learning procedure in each condition. Fig 4 reports the correlations between these variables and expression in the different conditions. First, we can observe that the individual correlations between these variables and gene expression are often much lower than that of the complete model, which was somewhat expected if we consider that the measured gene expression level is actually the result of several regulatory mechanisms. In accordance with the above permutation experiments, we also observe that for *P. falciparum*, *P. berghei*, *T. gondii*, as well as for *Drosophila* and *C. elegans* development series, the correlation between variables and expression fluctuates along the conditions, while for the other series correlations are steadier. For *P. falciparum* we can even observe a sinusoidal behavior, highly reminiscent of the sinusoidal pattern of expression observed in the different studies monitoring gene expression during the erythrocytic cycle [66–68]. We can also observe that the locations of cLREs are diverse, depending on species and conditions. Not surprisingly, in human the CG frequency in the region around the TSS is by far the most predictive variable. In the following, we assigned the variables to six different genomic regions: distal and close promoter, center region, 5'UTR, gene body, and whole region (see Materials and methods for details about how variables were associated with one of these locations). Then, for each variable, we computed a usage statistics that summarizes its importance in the learned models (see Materials and methods). Fig 5 reports the relative importance of the 6 regions in the different conditions (see also S6 Fig which summarizes the variable usage in the different conditions). Some interesting trends emerge from this analysis. For example, we can observe the importance of distal promoter regions in *P. falciparum* compared to all other species. In human, *Drosophila* and *C. elegans*, center and/or 5'UTR variables are important, especially for tissues. Interestingly, these variables seem less important in gonads and at early time points of *Drosophila* and *C. elegans* development, but take increasing importance along the course of the development of these species. Similarly, for *P. falciparum* the role of promoter variables decreases along the phases of the erythrocytic cycle.

Several studies have shown that regulatory elements can be conserved across high phylogenetic distances. Notably, the role of CpG islands in transcription regulation is likely shared in the vertebrate taxon [36], and a recent study suggests that the enhancer regulatory code may be conserved across animals [69]. Hence, as a first attempt to assess the conservation of the identified cLREs along evolution, we gathered the most important variables identified in each species and conditions, and computed the correlations between these variables and expression in every species and conditions. Then, an unsupervised clustering was run to classify the conditions according to these correlations (see Fig 6a). As we can see, conditions can be perfectly classified on the basis of these correlations (all conditions related to the same species group together). Interestingly, *P. falciparum* and *P. berghei* conditions also group quite clearly together and, with a less clear signal, with *T. gondii* conditions, while the remaining of the groupings does not seem to be in accordance with the phylogenetic tree of Eukaryotes. Looking at the correlation conservation of each variable individually gives a more precise view of this general trend (Fig 6b). Several variables are correlated with expression for both *P. falciparum* and *P. berghei* (*e.g.* ATA [-1196,-126]). At the *Apicomplexa* level, the number of common variables is low but still present (*e.g.* TTT [-684,2000]). Similarly, some variables are common to *Drosophila* and *C. elegans*, and a few ones seem common to *Drosophila*, *C. elegans* and human (CG [-125,1196]).

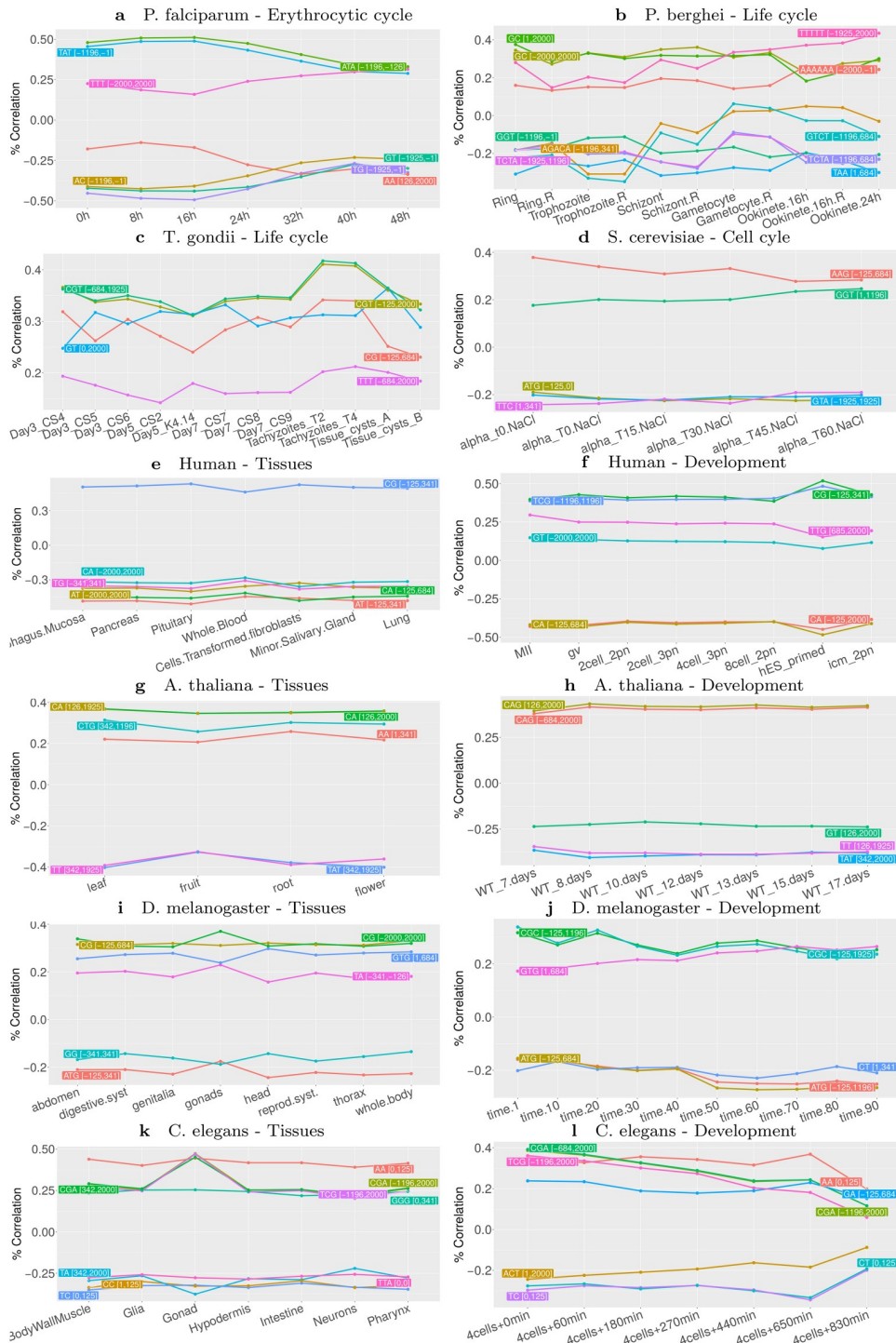

**Fig 4. Correlations between expression and k-mer frequency of the most important variables identified in the different species and conditions.** For each expression series, the 5 most important variables of each condition were identified, and their correlation to expression were computed for all conditions of the series. The name of the variables has been shortened for readability: for example the variable ATA [-1196,-126] is actually the frequency of k-mer ATA in region [-1196,-126]. Note that there are often more than 5 variables in these figures because the 5 most important variables may vary depending on conditions.

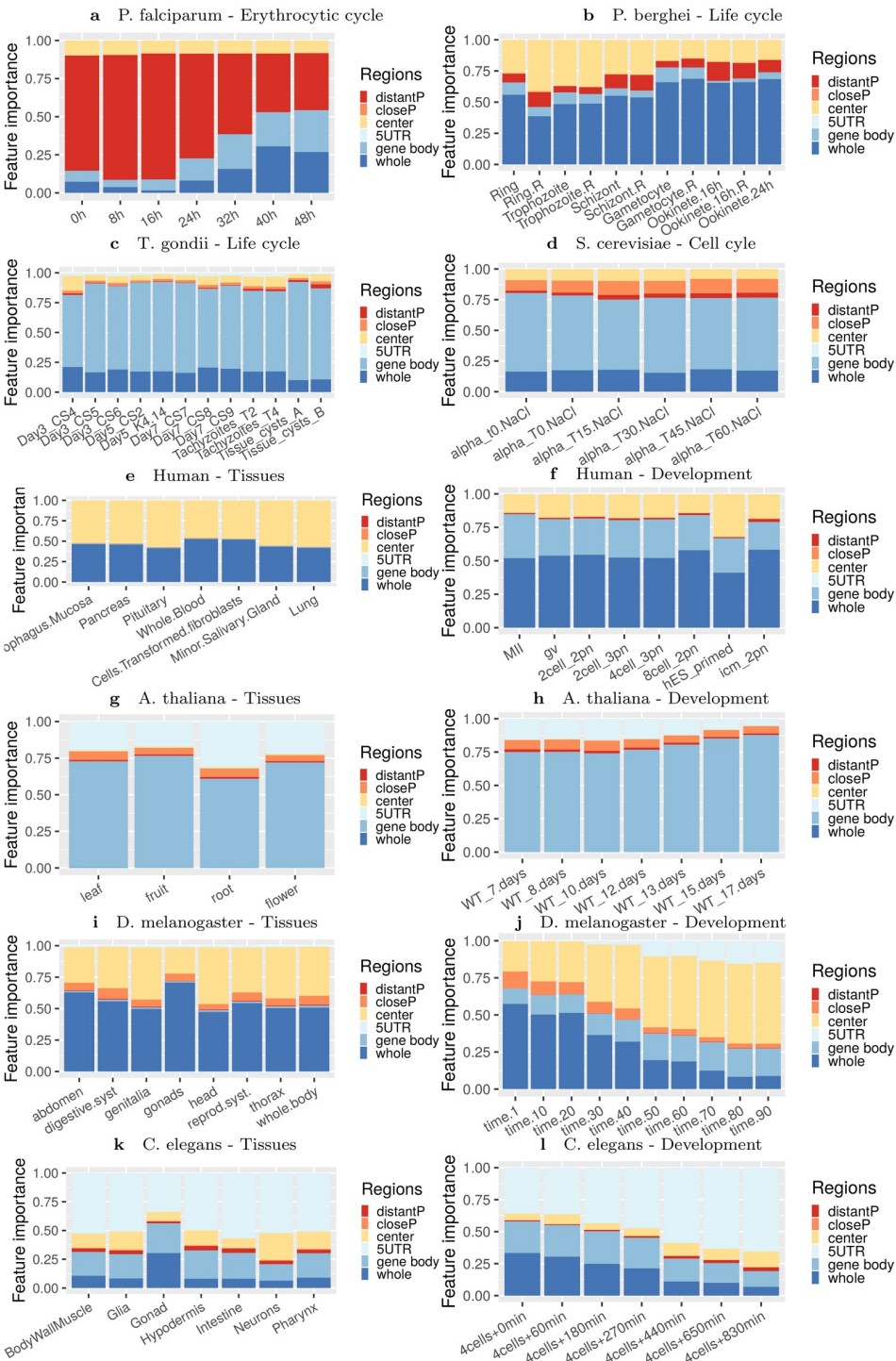

**Fig 5. Relative importance of promoter, untranslated and coding regions for predicting gene expression in different species and conditions.** For each condition, the 30 most important variables of the model were identified and a usage statistic reflecting the importance of the variables for the prediction was computed (see Materials and methods). Then, each variable was associated with one gene region (6 different regions were considered: distal and proximal promoters, center, 5'UTR, gene body, or whole; see Materials and methods), and the usage statistics of the variables that belong to the same region were cumulated.

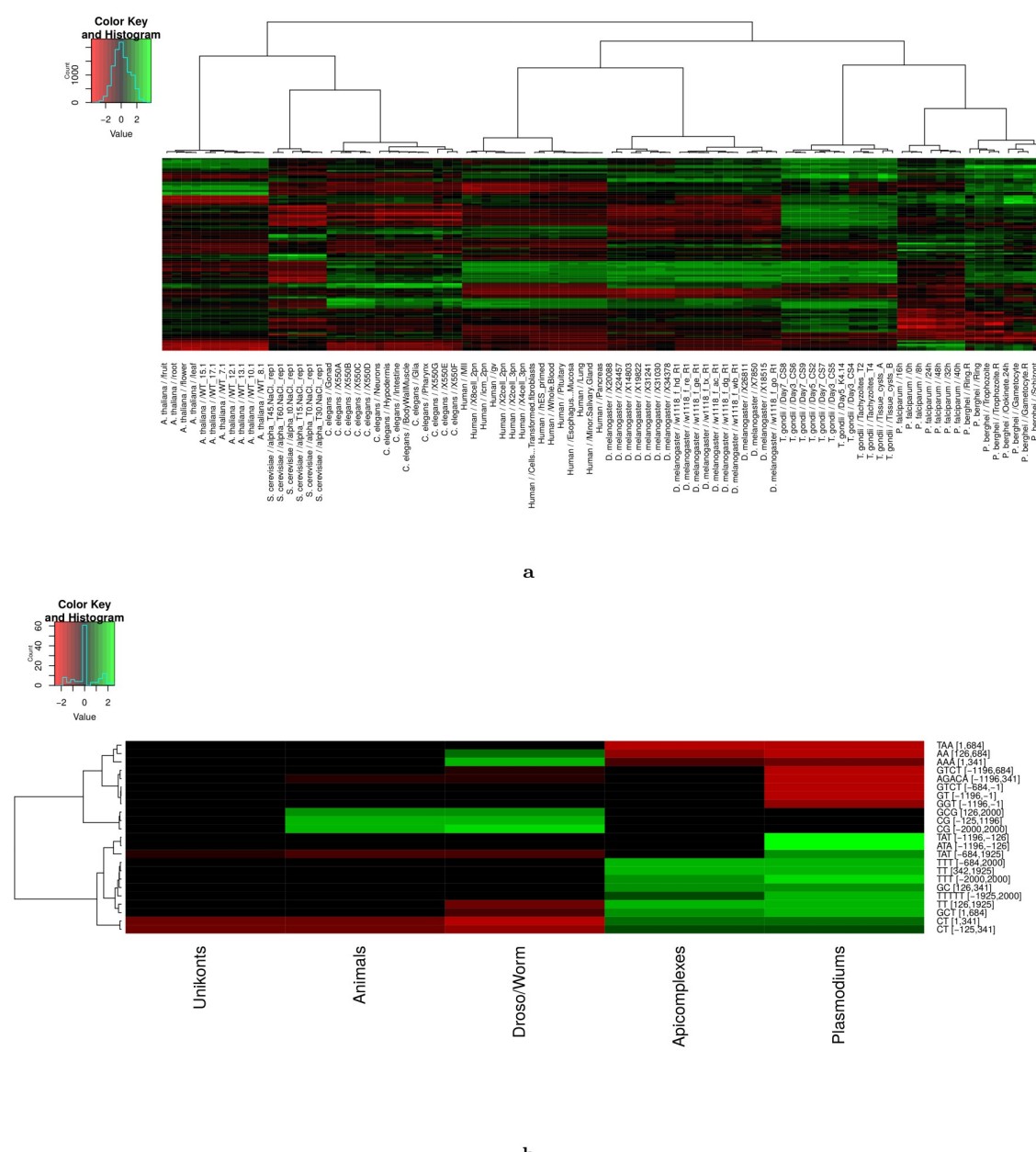

**Fig 6. Conservation of long regulatory elements along evolution.** The 10 most important variables of each species and conditions were identified and collected, and their correlations with expression were computed for every species and conditions. Correlations were then normalized by conditions (i.e. correlations were divided by the standard deviation of all correlations computed for the condition) to get the same range of values for each condition. **a** A hierarchical clustering (Ward's criterion) was run to classify the conditions according to these correlations. **b** The heatmap represents the variables whose correlation with expression is conserved at the level of at least one of five different taxa. The variables that do not show conservation of correlation at any of the 5 taxa have been removed for readability.

## cLREs are associated with highly dynamic profiles along *P. falciparum* life cycle and with specific GO terms

As explained above, the accuracy of the predictions is especially high for *P. falciparum*, culminating to 74% in the first stages of the erythrocytic cycle. For comparison, we trained several deep learning models (CNNs) on the same *P. falciparum* data. We used for this an architecture

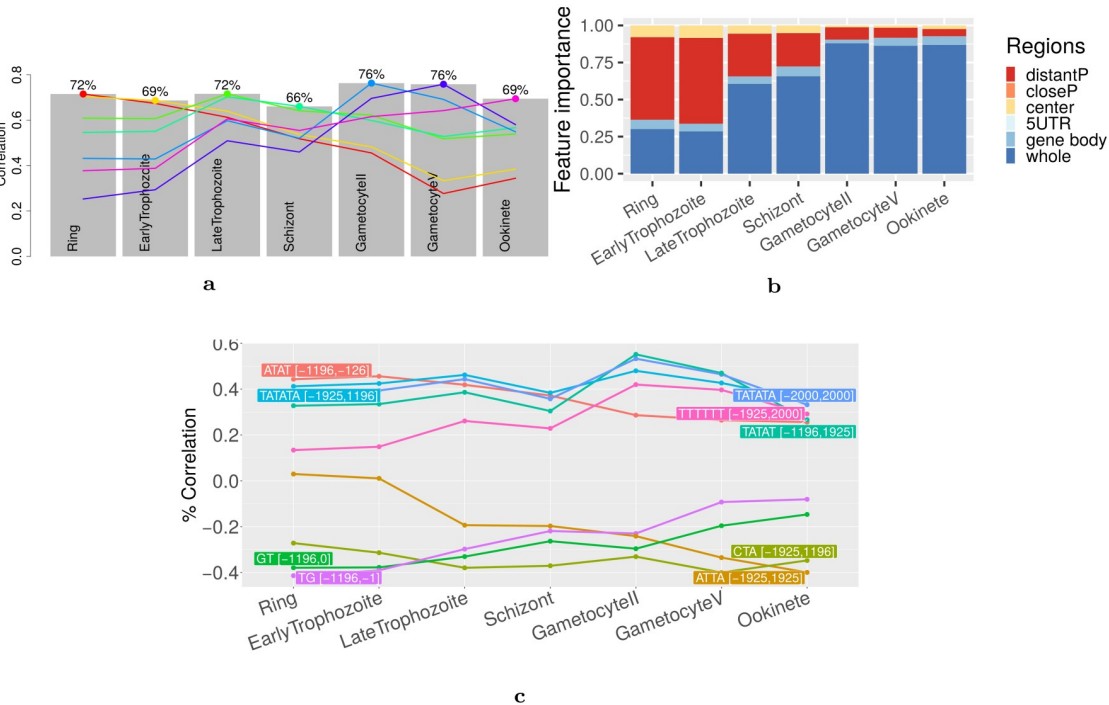

**Fig 7. Importance of cLREs along the whole life of *P. falciparum*. a** Grey charts represent the accuracy, measured as the correlation between predicted and observed gene expression, of the models learned on different phases of *P. falciparum* life cycle. Colored curves summarize the accuracy of a model learned on a specific phase when used to predict gene expression of other phases. **b** Estimate of the importance of upstream, downstream, center and whole regions for predicting gene expression in the different phases. **c** Correlations between expression and k-mer frequency of the 5 most important variables identified at each phase. Because the most important variables vary depending on conditions, the total number of variables is > 5 in this figure.

similar to that used in DeepSea [25] (see Materials and methods and S7 Fig). The accuracy achieved in these experiments is lower than that obtained with DExTER, but the same dynamic behavior is observed, indicating that CNNs can also capture, at least partially, cLRE effects (see section Discussion for possible explanations of the differences observed between CNNs and DExTER). Although cLREs can be identified in all studied Eukaryotes, the higher accuracy in *P. falciparum*, along with its dynamic behavior suggests that cLREs are particularly important for gene expression regulation in this species. Hence, *P. falciparum* appears as a model of choice for studying the regulatory mechanisms associated with such sequences.

To measure the extent to which cLREs control gene expression along the whole life cycle of *P. falciparum*, we ran an analysis of the data of Lopez-Barragan et al. [70] that measures gene expression in sexual and asexual stages of the parasite. Results are summarized in Fig 7. They are globally concordant with those achieved on data exclusively targeting the erythrocytic cycle, with accuracy above 70% in several stages. What is striking, however, is the highly dynamic behavior of the regulation process, something already observed on *P. berghei* life cycle (see Fig 2b): a model with high accuracy on gametocytes has very poor accuracy in asexual stages (particularly in ring stage), and reciprocally. This can be also observed by the high fluctuations of correlation between variable frequencies and gene expression (Fig 7c). Among the best variables identified by DExTER at the different stages, several ones are similar to those identified in the data of Otto et al. [51] (for example ATA and TG on upstream sequences, or the T repeat on whole sequences). Some others seem more correlated with expression in sexual stages than in asexual stages. For example TATAT in [-1196,1925] fluctuates between 30% and

50% correlation, while ATTA[-1925,1925] fluctuates between 0% and -40% correlation. On the whole, upstream variables seem highly important at the beginning of asexual stages, but much less in gametocytes and ookinetes (Fig 7b).

We next used the GSEA method [71] to analyze some of the variables that show the highest correlation with expression in different phases. Interestingly, genes enriched for specific variables are also associated with specific GO terms (see S8 Fig). For example, genes with high ATA frequency on upstream sequences are associated with high expression in early phases of the erythrocytic cycle and are involved in translation. Genes with high TTT frequency on the whole sequence are highly expressed on later time points and are involved in transport regulation. Similarly, genes with low AA on downstream regions are associated with high expression on late stages and are involved in different metabolic processes. Finally, genes with high TATATA frequency on the whole sequence are more expressed in gametocytes and are involved in chromatin assembly.

## cLREs are associated with transcriptional and post-transcriptional signals in the *P. falciparum* intraerythrocytic cycle

We next analyzed more closely the timing associated with the cLREs identified in the intraerythrocytic cycle (IEC). Fig 8a shows the 8 most important variables identified in each stage of the IEC (representing a total of 10 different variables). Left and right heatmaps present the variables with higher correlation in early (0h-16h) and late (24h-48h) stages of the IEC, respectively. In accordance with results presented in Fig 5, upstream variables are more correlated to expression in early time points, while downstream and whole variables show more correlation in late time points. Next, we estimated the strand specificity associated with each variable. For this, we computed the frequency of the corresponding k-mer in the identified region on the plus strand (as it is done in step 1 of DExTER) and on the minus strand, and we compared the correlations between these two frequencies and expression. Variables for which correlations differ between strands are considered as strand-specific (see Materials and methods for details). In Fig 8a, strand specificity is represented with a color code that goes from blue (no strand specificity) to orange (high strand specificity). Interestingly, all upstream variables show little or no strand specificity, while three among the four variables with highest correlation in late stages are strand-specific.

The absence of strand specificity in upstream cLREs and its presence in downstream/whole cLREs suggest that upstream and downstream cLREs may be involved in transcriptional and post-transcriptional regulation mechanisms, respectively. To assess this point, we analyzed the data of Painter et al. (2018) [15], where the level of nascent transcription and stabilized mRNA along the IEC are measured in parallel. We ran DExTER on these two types of data and for each available time point, and identified the 8 most important variables on each condition. Among the 15 different variables, 4 are clearly more correlated with nascent transcription than with stabilized transcript levels, while 5 others are more associated with stabilized transcripts (Fig 8b) (the remaining variables cannot be clearly associated with one or the other type of data, see Materials and methods). Remarkably, all variables associated with nascent transcription are both upstream and non strand-specific, while variables associated with mRNA stabilization are downstream and strand-specific.

This raises a question about the nature of the few variables that span the whole sequence. This is typically the case of the variable TTT[-2000,+2000] on IEC (see Fig 4a) and of variable ATTA[-1925,1925] on the whole parasite life cycle (see Fig 7c). We thus measured separately the correlation with expression and the strand specificity of these variables upstream and downstream of the ATG, using time point 48h and the ookinete stage as references. First, for

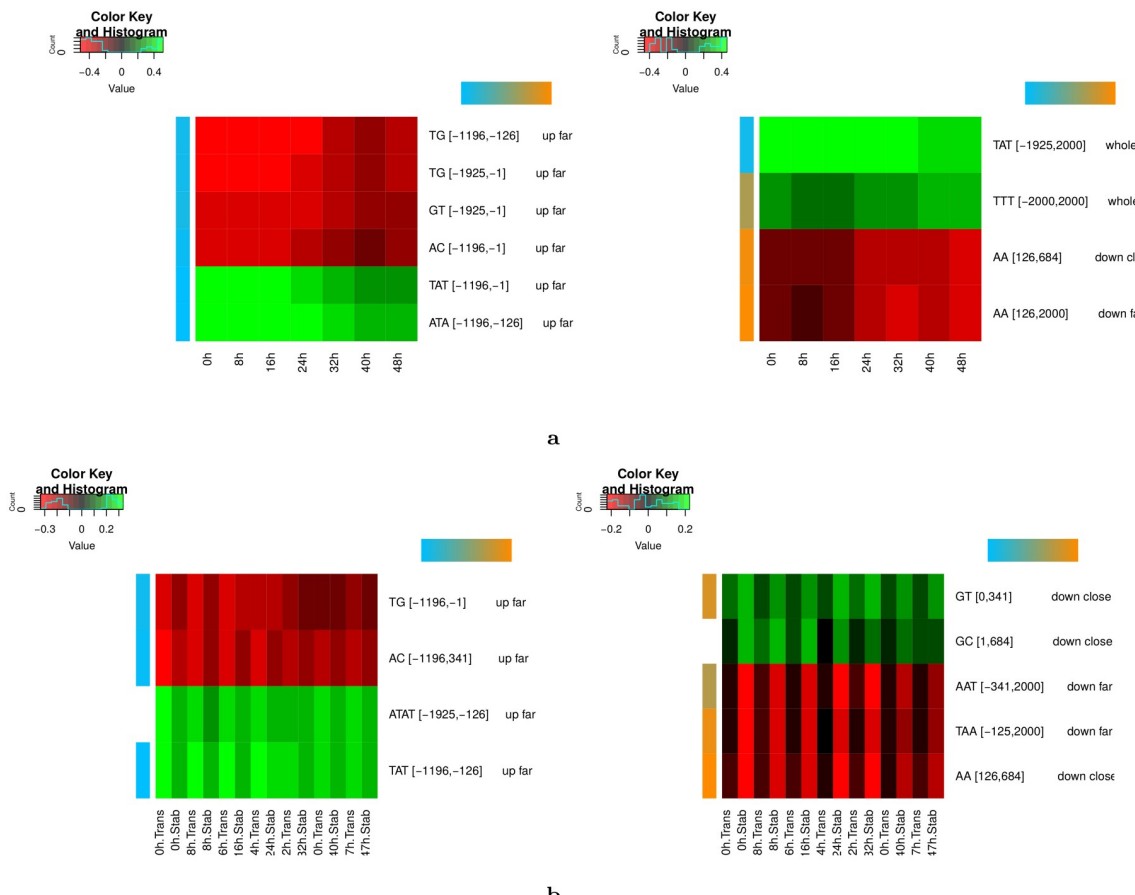

**Fig 8. Strand specificity of cLREs and links with post-transcriptional signals in *P. falciparum* intraerythrocytic cycle. a** Heatmaps of correlations between gene expression and most important features identified at each time point of Otto et al. (2010) data. The left heatmap corresponds to features with higher correlation in early time points (0h—16h), while the right heatmap corresponds to features with higher correlation with late time points (24h—48h). The strand specificity of each variable is represented with a color code that goes from blue (no strand specificity) to orange (high strand specificity). **b** Heatmaps of correlations between gene expression and most important features identified at each time point of Painter et al. (2018) data. The left heatmap corresponds to features with higher correlation with transcription data, while the right heatmap corresponds to features with higher correlation with stabilization data.

both variables, the correlation with expression is higher for the whole sequence (34% and -37% for TTT and ATTA respectively) than for the upstream (19% and -26%) or downstream (26% and -30%) sequences solely. Second, the strand specificity of these variables is low for upstream sequences (the correlation with expression is almost the same when the variables are computed on the plus or on the minus strand) but high for downstream sequences: the correlation of ATTA with gene expression drops to 8% only, and the correlation of TTT with expression is inverted (-27% on the minus strand *vs.* +27% on the plus strand). Hence, a possible hypothesis for these whole-sequence variables would be that they actually involve two LREs acting coordinately: one upstream LRE associated with transcriptional regulation mechanism, and one downstream LRE likely associated with post-transcriptional regulation. Because the two LREs involve the same k-mer and act in a coordinate way (they are both either positively or negatively correlated with expression) they appear as a single variable in the DExTER analysis.

## Links with histone modifications and variants in *P. falciparum*

Read et al. [63] have shown that gene expression in *P. falciparum* can be predicted with rather good accuracy from various epigenetic marks. Notably, histone variant H2A.Z, and histone modification H3K9Ac and H3K4me3 in promoters and gene bodies appear to be among the most predictive marks for expression. Hence, we sought to assess whether some of the predictive variables identified by our approach could actually be related to these specific marks. To do so, we used the data of Bartfai et al. [72] to compute the H2A.Z, H3K9Ac, and H3K4me3 signals upstream and downstream the AUG codon of every gene, and we ran DExTER to predict these data instead of gene expression. Globally, prediction accuracies are lower than for gene expression. Only H2A.Z and H3K9ac downstream signals can be predicted with accuracy around 60%, but without reaching the > 70% accuracy achieved for gene expression (see S9 Fig). Analysis of the most important variables of the H2A.Z and H3K9ac downstream models shows that several variables identified for gene expression are also singled out when predicting these histone marks (S10 Fig), but none of these variables seems more correlated with histone marks than with gene expression.

## Links with antisense transcript levels in *P. falciparum*

It has been shown that *P. falciparum* produces numerous antisense RNAs. Former studies estimate that around 25% of coding genes have significant antisense levels during blood stage development [73]. We thus asked whether cLREs could be associated with the regulation of antisense RNA level also. For this, we used DExTER to predict the expression level of the data of Siegel et al. [73], who measured independently the level of sense and antisense transcript of all coding genes. For sense transcription data, we used 4000 bp sequences centered on the start codon (AUG) as in our previous analyses. For antisense data, we used the same sequences, as well as 4000 bp sequences centered on the stop codons. Results are summarized in S11 Fig. For sense transcripts, we obtained similar results to those obtained on the data of Otto et al. [51] and Lopez-Barragan et al. [70]. For antisense transcripts however, the prediction accuracy was significantly lower. Different reasons may explain this discrepancy but one possibility would be that the very low expression level of these transcripts actually hinders their proper quantification. Interestingly however, we obtained better results with sequences centered on the stop rather than on the start codon for these transcripts, suggesting that antisense RNAs may possess their own promoter sequences, and that LREs can also be involved in their regulation.

## Links with genome AT content

One striking observation is that *P. falciparum* and *P. berghei*, the two organisms for which cLREs have the strongest links with gene expression—both in terms of predictive performance and dynamic behavior—, are also the organisms with the most AT-biased genomes. One question is then to know if the importance of cLREs for the regulation of expression could be linked to the genome AT-content. To answer this question, we ran DExTER on gene expression datasets of *Dictyostelium discoideum* and *Tetrahymena thermophila*, two other organisms selected for the high AT content of their genome [74]. For these experiments, we analysed the RNA-seq data of time course development of Nichols et al. [75] and Zhang et al. [76]. Results achieved on these two species contrast strongly (see Fig 9). On *D. discoideum*, the accuracy achieved by DExTER is close to that obtained on the two *Plasmodium* species, and a dynamic behavior is also observed, with marked differences between models learned at the beginning and end of the development course. Conversely, for *T. thermophila* the accuracy stays around 50% and no dynamic behavior is observed. Hence, if the best predictive performances of this study have been obtained with high AT-content genomes (*P. falciparum*, *P. berghei* and

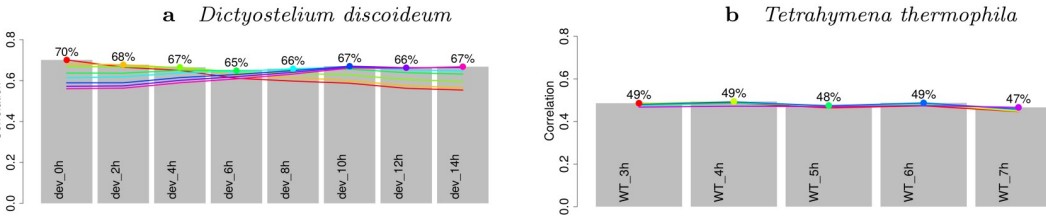

**Fig 9. DExTER accuracy on *D. discoideum* and *T. thermophila*, two other genomes with high AT content.** Grey charts represent the accuracy (y-axes), measured as the correlation between predicted and observed gene expression, of the models learned on different conditions (x-axes). Colored curves summarize the accuracy of a model learned on a specific condition (identified by a big dot of the same color) when used to predict the other conditions of the series.

*D. discoideum*), it seems that there are also AT-biased genomes for which the link between cLREs and gene expression is not as strong. This also indicates that DExTER prediction power is not particularly biased towards AT-rich genomes.

## Reporter assay validates a LRE controlling gene expression in *P. falciparum*

Finally, to validate our approach and demonstrate the importance of LREs in *P. falciparum*, a GFP reporter assay was performed. One of the most important variables identified by DExTER is the ATA frequency in region [-1196,-126]. This variable by itself links gene expression and k-mer frequency with nearly 50% correlation at ring stage parasites (Fig 4a). For our analyses, we built two chimeric promoters on the basis of the PF3D7_0913900 promoter. We chose this gene because although the frequency of ATA in region [-1196,-126] is low, it has high ATA content in a very short region [-483,-128]. As expected, this gene, nonetheless expressed in asexual stages is not highly expressed in ring stages [51, 70]. We built a first chimeric promoter containing 3 repetitions of the region with high ATA frequency [-483,-128] (see Fig 10a). As a control, we constructed a second promoter replacing two of these repetitions by a region of low ATA frequency (see Fig 10a). Both of these chimeric promoters were of similar length and GC content. Then we used the DExTER model learned at 8h (ring stage parasites) of the erythrocytic cycle [51] to predict the transcriptional activity of both promoters when associated with a GFP gene. DExTER predicted a higher activity for the promoter with 3 repetitions than for promoter with 1 repetition. To validate these predictions, each promoter driving the expression of a reporter GFP gene was integrated in the genome of *P. falciparum* using the CRISPR/Cas9 technology ([77]; see Materials and methods). The GFP expressing cassette was

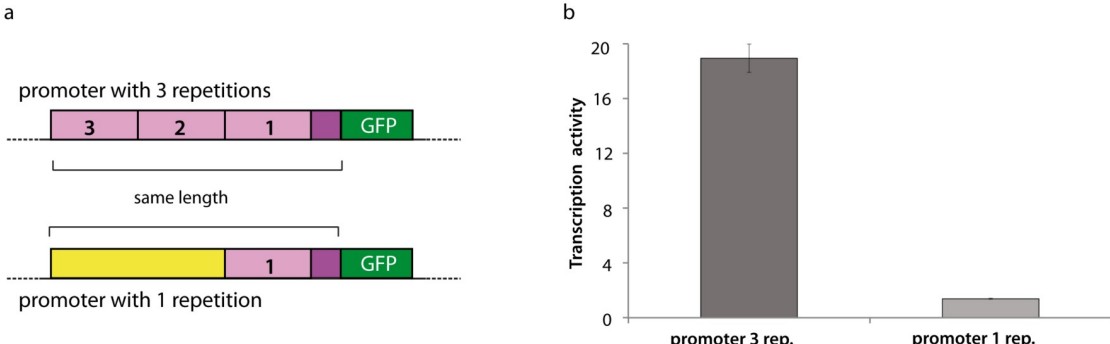

**Fig 10. In vivo experimental validation in *P. falciparum*. a** Schematic of the chimeric promoters used in our report assay to monitor promoter activity. **b** Transcriptional activity quatification by qPCR analysis of RNA collected at ring stages parasites. Here, one representative transgenic parasite clone. See Materials and methods for details.

inserted into the locus Pfs47, shown to be dispensable for parasite blood stage development and previously used for transgene expression in asexual stages of GFP and other genes [78, 79]. The transcriptional activity of both promoters was measured by qPCR analysis of RNA collected at ring stages for each transgenic parasite line. Transcriptional activity from the promoter with 3 repetitions of the high ATA frequency region was 10-fold higher than that of the promoter with 1 repetition (Fig 10b). As a control, we also measured the transcriptional activity of one housekeeping gene (fructose-biphosphate aldolase PF3D7_1444800) and observed very similar activity in the two genetically modified parasites (S12 Fig).

## Discussion

Gene expression in eukaryotes is orchestrated at different levels and by different mechanisms to ensure the wide variety of responses associated with the different cell types, stages and conditions. Besides traditional short TFBS, long regions with specific nucleotide compositions may constitute another type of regulatory elements [80, 81]. While several *in silico* approaches exist for characterizing short TFBS, to our knowledge, no methods dedicated to LREs have been proposed so far. We present in this paper a computational approach specifically designed to characterize cLREs correlated with gene expression. Applied to various genomes and expression data, our method revealed that LREs seem to exist and to be active in a wide range of species and conditions. The nature of these regions greatly varies between species and, in some cases, between conditions, but, surprisingly, they seem to control a substantial part of gene expression in all studied organisms.

One striking observation is the distinct regulation dynamics in the different species. Regulation with cLREs appears to be highly dynamic in *Plasmodium*, *T. gondii*, and *D. discoideum*. This dynamic behavior may appear somewhat surprising as it is dictated by the sequence of the genome, which is intrinsically static. This is however simply the reflect of the interplay that occurs in cells between cis-elements—which are statics—and DNA-binding molecules, whose nature and quantity vary depending on conditions. Thus, the presence of a specific cis-element in the vicinity of certain genes may for example induce high expression in conditions where the associated DNA-binding molecule is active, while being inoperative in other conditions. In our framework, this is translated by the fact that the models learned at different conditions do not use the same variables (cis-elements). Hence, a model learned at a given condition may have poor accuracy on other conditions. In multicellular organisms however, these mechanisms seem to be much more static when different tissues are compared. One hypothesis could be that cLREs in these species are used to fit a kind of rough, basal, transcription level for each gene. Adjustments to these basal levels could then be done in a tissue dependent way by other mechanisms that do not involve LRE. Interestingly, contrary to what is observed in tissues (with the exception of gonads), *Drosophila* and *C. elegans* embryo development also show a dynamic behavior that is accompanied by a switch of the most important cLRE positions. The gene body and the whole region seem more important at early time points, but they gradually lose their importance along the course of development in favor of central or 5'UTR regions.

In *P. falciparum*, cLREs seem very important at every phase of the life cycle, and especially in IEC. Experiments with CNNs corroborated these results, although the accuracy achieved with these models was lower than with DExTER. Several reasons may explain the difference of accuracy, one reason being that the CNN architecture used here, which is classically used in regulatory genomics to identify TFBS motifs, may not be the best architecture to capture cLRE features. Other architectures have indeed been proposed to predict transcription from CAGE data [28, 29]. Hence, from a purely predictive perspective, it seems that some work is needed

to propose an architecture capable of fully capturing cLREs with CNNs. On the other hand, we must stress out that controlling what is exactly learned by CNNs is a difficult task [33]. When used for modeling gene expression, these models likely capture a mixture of regulatory elements, including traditional TFBS motifs, LREs, and potentially many other kinds of regulatory elements. Because disentangling all these effects seems hazardous, we believe that direct approaches like DExTER constitute better alternatives than CNNs for studying the specific effect of cLREs.

Our study also suggests that cLREs are associated with transcriptional and post-transcriptional regulation in *P. falciparum*. cLREs associated with nascent transcription are both upstream the initiation AUG and non strand-specific, suggesting their involvement in transcriptional regulation mechanisms, while cLREs associated with mRNA stabilization are downstream and strand specific, pointing to post-transcriptional regulation mechanism. Several studies provided evidence for a control of gene expression at the post-transcriptional RNA level in this parasite. Lack of coordination between active transcription and mRNA abundance has been reported in *P. falciparum* [82, 83] and bioinformatics analysis have indicated that a significant percentage of the *Plasmodium* genome encode RNA-binding proteins (4-10%) [84, 85].

Finally, in vivo analysis of the promoter activity of a chimeric DNA fragment showed higher transcriptional activity when the region was enriched in one of the cLREs identified by DExTER as positively associated with RNA levels. These results should now be confirmed by additional experiments in order to precisely assess the role and importance of this LRE in *P. falciparum* biology. Notably, it would be interesting to replicate these experiments in varying the number of repetitions of the high ATA frequency region, as well as to monitor the transcriptional activity of these chimeric promoters along the entire erythrocytic cycle. Next, it would be also interesting to investigate the molecular mechanisms underlying the role of this LRE in gene regulation such as protein recruitment, nucleosome occupancy alteration or modulation of epigenetic marks. Indeed, the apparent variety of cLREs likely implies heterogeneous mechanisms of gene regulation. Obviously, the exact nature of these mechanisms remains to be investigated and constitutes the main question raised by this study. cLREs may constitute "loose" binding sites for certain DNA or RNA binding proteins as proposed by Quante and Birds [45], that may regulate for instance nucleosome occupancy [40], 3D genome architecture [42] and/or alternative splicing [46]. DExTER provides a tool that will help design dedicated experiments aimed at better characterizing the contribution of cLREs in these processes.

Another question raised by this study is the reason for the apparent higher importance of LREs in *Plasmodium*, compared to other species. This might be linked to the paucity of TFs [8, 9] but also to the scarcity of distant regulatory sequences (enhancers) identified in *Plasmodium*, despite some work performed recently [86, 87]. The potential regulatory mechanisms of short tandem repeats [88] may also explain the prominent role of LREs in *Plasmodium* gene regulation. For example, despite the low abundance of TFs, the LREs may alter TFs binding, increasing their regulatory potential. Finally, LREs may also modulate expression changes through altering nucleosome positioning [89]. In line with this question is the apparent difference in the relative importance of distal promoter regions between *P. falciparum* and *P. berghei* (Fig 5). A closer look (Fig 4 and S6 Fig) shows that the importance of this region in *P. falciparum* is mostly brought by the variable TAT (or ATA), whose frequency in distal promoter region is highly correlated with expression in early time points of the intraerythrocytic cycle. Although this variable is also correlated with expression in *P. berghei* (the correlation is above 20% for most time points), it is not among the best predictive variables for this organism. This point may be linked to the fact that, although the AT content of the genes of *P. berghei* and *P.*

*falciparum* are almost the same (76.22% and 76.25%, respectively), the AT content of intergenic regions is more skewed in *P. falciparum* (85.72% vs. 80.05%) [74].

Another interesting observation is the prevalence of AT-rich k-mers in the identified cLREs in *P. falciparum* genome. This may be somewhat surprising in a genome composed of more than 80% A+T, as any region is expected to be enriched for such k-mers. While this assertion is globally true, the frequency of a specific AT-rich k-mer in a specific region can nonetheless fluctuate between genes, and our study shows that these fluctuations are linked to gene expression. Moreover, we also showed that all AT-rich cLREs have not the same correlation profile with gene expression (see Fig 7c). For example, ATAT [-1196,-126] has high correlation with expression in the IEC but lower correlation in gametocyte and ookinete stages, while TATAT [-1196,1925] shows moderate correlation in the IEC and higher correlation in gametocyte. Similarly, while the above mentioned AT-rich cLREs are positively correlated with expression, ATTA [-1925,1925] appears to be negatively correlated with expression, especially in gametocytes and ookinetes. Hence, AT richness per se cannot be associated with a standardized global response but it seems on the contrary that *P. falciparum* has developed a subtle regulatory vocabulary largely based on these two nucleotides which, depending on the region and the exact k-mer, may generate different responses.

## Materials and methods

### The DExTER method

**Step 1—Feature extraction.** We developed a procedure to identify pairs of (k-mer,region) for which the frequency of the k-mer in the region is correlated with gene expression. Starting from a 2-mer in the whole sequences, the procedure alternates two steps.

- The first step is the **segmentation** step (magenta arrows in the exploration graph of Fig 1). For this, sequences are first segmented in different bins defined from the alignment point (anchor). We used 13 bins in our experiments. The size of the bins are determined with the polynomial $(x + a)^3$, with $x$ being the rank of the bins with respect to the anchor (the bin centered on the anchor has rank 0, while bins immediately on the left and right of this bin have rank 1, etc.). $a$ is a parameter determined automatically by the procedure in order to cover the whole sequences with the required number of bins (here 13) in the best possible way. With this method, bins close to the anchor are shorter than bins away from this point. When the binning is done, a lattice representing different regions that can be constructed from these bins is computed (see Fig 1). The top of the lattice represents the whole sequence, while lower nodes represent smaller regions. At each node, the correlation between the 2-mer frequency in the associated region and gene expression is computed, and the region with highest correlation is identified. If this correlation is sufficiently higher than the correlation associated with the top node (whole sequences) the region is selected. This procedure is resumed on all non-overlapping regions until the whole sequences are covered or the remaining correlations are lower than the correlation of the top node.

- Every identified region is then investigated for an **expansion** step of the 2-mer (green arrow in the exploration graph of Fig 1). Here, the goal is to identify (k+1)-mers whose correlation on the identified region is higher than the original k-mer. For this, the 8 possible (k+1)-mers obtained by concatenating a nucleotide on the left or right of the k-mer are constructed. The correlations between gene expression and the frequency of these new k-mers in the region are computed, and k-mers that improve correlation are identified.

The whole procedure (segmentation + expansion) is resumed iteratively, until no improvement is observed. Then a new exploration starting from a different 2-mer is ran until every 2-mer has been explored. At each step of these explorations, regions and k-mers that improve correlations over the previous step (*i.e.* all nodes of the exploration graph of Fig 1) are stored and form the list of variables returned at the end of the procedure.

**Step 2—Feature selection and learning.** Once all potential variables have been extracted, a regression model is learned and the best variables are identified. If only one gene expression data set is available, variable selection (Eq 1) is performed using the LASSO (Least Absolute Shrinkage and Selection Operator) [50]: by penalizing the absolute size of the regression coefficients (l1-norm), the LASSO drives the coefficients of irrelevant variables to zero, thus performing automatic variable selection.

If several gene expression data are available for one species (as it is the case in the paper), we make use of multitask learning so that all models are learned simultaneously with a global penalization. Multitask learning exploits the relationships between the various learning tasks in order to improve inference performance. Here, in order to stabilize variable selection, we encourage that each feature is either selected in all samples or never selected. The group LASSO [90] is naturally suited for this situation. In particular, if the feature extraction step has identified the same k-mer in similar but slightly different regions in the different data, group LASSO encourages the selection of a common region for all models.

## Estimation of model accuracy

Once the features have been extracted and a model has been trained, its accuracy is evaluated by computing the Pearson correlation between predicted and observed expressions on several hundred genes. To avoid any bias, this is done on a set of genes that have not been used in the two previous steps. In our experiments, we used 2/3 genes for training, and 1/3 genes for testing. Note that the same sets of training and testing are used for all conditions of the same series. This avoids any optimistic bias when the accuracy of a model learned in one condition is evaluated on the other conditions in the permutation experiments.

## TF binding motifs

We used the TF binding motifs from the JASPAR 2020 CORE non-redundant databases Fungi, Vertebrates, Plants and Insects for identifying potential TF binding sites in *S. cerevisiae*, human, *A. thaliana* and *Drosophila* sequences, respectively. For *P. falciparum* we used the 23 motifs identified in Campbell et al. [64] and downloaded from the MEME suite [91]. For each organism, sequences were scanned with FIMO [92] with option `--text` and `--thresh 0.001`. Then, for each motif and sequence, we kept the minimal p-value obtained for this motif in this sequence, and take the negative of the logarithm of this p-value as motif score. Missing values (*i.e.* motif with p-value above the 0.001 threshold for a sequence) were replaced by value 1.

## Convolutional neural networks

We build a convolutional neural networks using the `keras` implementation (Chollet, F. et al. Keras, 2015; https://keras.io) and with an architecture similar to the architecture proposed in DeepSea [25], *i.e.*:

- convolution layer (32 kernels, window size 8, step size 1, dropout 20%),

- max pooling layer (size 2),

- convolution layer (64 kernels, window size 8, step size 1, dropout 20%),

- max pooling layer (size 2),

- convolution layer (64 kernels, window size 8, step size 1, dropout 20%),

- max pooling layer (size 2),

- dense layer (64 kernels, dropout 50%),

- flatten layer,

- dense layer (1 kernel).

In each kernel we used the activation function `relu` and the `glorot_uniform` initializer. The `adam` optimizer was used to minimize the mean squared error (MSE) loss function. 5% training sequences were used as validation sequences. We defined an early stopping condition on the MSE with a patience of 2. As for DExTER, 2/3 genes were used for training and the remaining 1/3 was used to measure the correlation between predicted and observed expression.

### Measure of variable importance

We devised an *ad hoc* procedure based on LASSO penalty and model error for measuring the importance of the different variables of a model. Given a penalization constraint λ, the LASSO procedure searches the model parameters that minimize the prediction error (MSE) subject to the constraint. In practice, a grid of constraints of decreasing values is initialized, and a model is learned for each value. The result is a series of models with increasing number of parameters. To identify the most important variables of a model in a given condition, we took the model with 15 parameters and estimated the importance of each of the 15 variables in the following way. Given a variable $X$, its importance was estimated by the MSE difference between the complete model and the model obtained by setting $\beta_X$ to 0.

### Variable locations

Variables were assigned to 6 gene regions: distal and close promoters, central region, 5'UTR, gene body, and the whole region. For 5'UTR and gene body, the region boundaries were defined on the basis of the median size of the annotations found for each species. Here, the gene body region refers to the genomic region downstream of 5'UTR (it can potentially include introns). Note that for *P. falciparum* and *P. berghei*, the sequences are aligned on the AUG instead of the TSS, so the 5'UTR is defined upstream point 0. For regions different from 5'UTR and gene body (*i.e.* distal and close promoters, central and whole regions) boundaries were not based on existing genome annotations and the same values were used for all species. Table 1 reports the boundaries used for defining all gene regions in the different species:

We used the Jaccard index for assigning each variable identified by DExTER to the gene region that most resembles it. Namely, given a variable region $R1$, we searched for the gene region $R2$ for which the ratio $\frac{|R1 \cap R2|}{|R1 \cup R2|}$ is the closest to 1.

### cLRE conservation

The 10 most important variables of each species and conditions were identified and collected, and their correlations with expression were computed for every species and condition. For each variable, correlations were then normalized by conditions (z-score) to get the same range of values for each condition. Next, for each species and variable, we then identified and

**Table 1. Boundaries defining gene regions in the different species.**

| species | dist. prom. | close prom. | center | 5'UTR | gene body | whole |
|---|---|---|---|---|---|---|
| *P. falciparum* | [-2000,-300] | [-300,-50] | [-500,500] | [-50,0] | [0, 1715] | [-2000,2000] |
| *P. berghei* | [-2000,-300] | [-300,-50] | [-500,500] | [-50,0] | [0, 1715] | [-2000,2000] |
| *T. gondii* | [-2000,-300] | [-300,0] | [-500,500] | [0, 454] | [454, 2000] | [-2000,2000] |
| *S. cerevisiae* | [-2000,-300] | [-300,0] | [-500,500] | [0, 43] | [43, 1014] | [-2000,2000] |
| Human | [-2000,-300] | [-300,0] | [-500,500] | [0, 86] | [86, 2000] | [-2000,2000] |
| *A. thaliana* | [-2000,-300] | [-300,0] | [-500,500] | [0, 131] | [131, 2000] | [-2000,2000] |
| *Drosophila* | [-2000,-300] | [-300,0] | [-500,500] | [0, 94] | [94, 2000] | [-2000,2000] |
| *C. elegans* | [-2000,-300] | [-300,0] | [-500,500] | [0, 26] | [26, 2000] | [-2000,2000] |

memorized the highest (in absolute value) normalized correlation found in any conditions of the species. We denote as $\rho_s^v$ the best correlation found for variable $v$ in species $s$. Then, at the level of the taxa, we used the minimum of the best correlations among all species of the taxa to measure the conservation of correlation of variable $v$ (see Fig 6, down). For example, for the Apicomplexan taxa, we took the minimum of $\rho_{Pf}^v$, $\rho_{Pb}^v$ and $\rho_{Tg}^v$ to assess the conservation of variable $v$ at the level of Apicomplexa.

## Strand specificity

The strand specificity of a given variable for a given condition was measured on the basis of the correlation between the frequency of the variable in the different genes and the expression of the genes in the condition. More precisely, two correlations were computed and compared: the correlation computed on the frequencies measured on the plus strand ($\rho_+$), and the correlation computed on the frequencies measured on the minus strand ($\rho_-$). The quantity

$$\frac{|\rho_+ - \rho_-|}{\max\left(|\rho_+|, |\rho_-|\right)}$$

was then used to measure the strand specificity of the variable. With this measure, variables for which correlations are approximately the same on the two strands have strand specificity around 0, while variables with high correlation differences between strands have higher strand specificity. Note that this quantity is meaningless for the few k-mers for which the reverse complement is equal to the original k-mer (*i.e.* CpG, GpC, TpA and ApT for dinucleotides), because in this condition the k-mer occurrences are the same for both strands.

## Transcription *vs.* stabilization variables

In Painter et al. (2018) [15], the authors measured separately the level of nascent transcription and stabilized mRNA along the erythrocytic cycle. We ran DExTER on each available time point for these two types of data, and identified the 8 most important variables on each condition and time point, giving us a total of 15 different variables. For each variable, we computed its correlation with nascent transcription and with stabilized mRNA at each time point, and we summed the absolute value of theses correlations separately. This gives us two quantities for each variable: $\rho_{v,trans}$ (resp. $\rho_{v,stab}$) is the sum of the absolute value of correlations of variable $v$ with transcription (resp. stabilization) at the different time points. Variables with $\rho_{v,trans} > \rho_{v,stab} + 0.3$ were associated with transcription, while variables with $\rho_{v,stab} > \rho_{v,trans} + 0.3$ were associated with stabilization. Variables with no clear differences were discarded.

## Gene expression data

- *P. falciparum*: RNA-seq data of the IEC were downloaded from the supplementary data of the original publication of Otto et al. [51] and were log transformed. Life cycle RNA-seq data [70] were downloaded from PlasmoDB. Log transformed FPKM data were used for these analyses. Transcription *vs.* stabilization data [15] were obtained from the paper Supplementary data 1 and log transformed. Sense and antisense expression data [73] were downloaded from the supp. mat. of the original publication (RPKM) and log transformed.

- *P. berghei*: RNA-seq data of the life cycle [52] were downloaded from PlasmoDB and log transformed.

- *T. gondii*: RNA-seq data of the life cycle [53] were downloaded from GEO (GSE108740), and the FPKM signal was log transformed. The data contains expression measures in tissue cysts, tachyzoites, and enteropithelial stages (3, 5 and 7 days post-infection).

- *S. cerevisiae*: RNA-seq data of NaCl stress response were downloaded from GEO (GSE89554) and log transformed. Only the conditions (alpha, NaCl) in *S. cerevisiae* were used for the analysis.

- Human: For tissues, RNA-seq data were downloaded from GTeX. We used the log of median TPM of 7 tissues for the analyses: Transformed fibroblasts, Esophagus—Mucosa, Lung, Minor Salivary Gland, Pancreas, Pituitary, Whole blood. For developmental series, we used data published in [60]. Expression data were downloaded from GEO (GSE101571) and log transformed.

- *Drosophila*: For tissues, we used the data of [55]. Expression data were downloaded from GEO (GSE99574, dmel.nrc.FB) and log transformed. Only the first biological repeat of each tissue was used in the analyses. For developmental series, we used the fly data produced in reference [56]. Data were downloaded from GEO (GSE60471, DM) and log transformed. 10 time points along the whole time series were analyzed.

- *C. elegans*: For tissues, we used data from the cell atlas of worm [57]. Data were downloaded from the Rdata file available on the cell atlas (http://atlas.gs.washington.edu/worm-rna/) and the log of the TPM of each available tissue were used for analyses. For developmental series, we used the data published in reference [58]. Data were downloaded from GEO (GSE87528, MA 20 strains) and log transformed. Only time points related to strain #550 were analyzed.

- *A. thaliana*: For tissues, we used the data published in ref. [61]. Data were downloaded from ArrayExpress (E-GEOD-38612, FPKM) and log transformed. For development, we used the series published in [62]. Data were downloaded from GEO (GSE74692, processed data) and log transformed. Only the first biological repeat of each time point of the wild type series were used for the analysis.

- *D. discoideum*: RNA-seq data were downloaded from GEO (GSE144888) and log transformed. We only used the developmental data series for this analysis.

- *T. thermophila*: RNA-seq data were downloaded from GEO (GSE104524) and log transformed. We only used the data of the wild type species for the analysis.

## Genome data

We analysed +-2Kb sequences centered on the TSS or AUG (except for *P. falciparum* and *P. berghei*, where the start codon is used, all genome sequences are centered on the most

upstream TSSs of the coding genes). We chose +-2kb because for *P. falciparum*, which constitutes the main targets of this work, and for several other organisms of this study (*P. berghei*, *T. gondii*, *S. cerevisiae*, *D. discoideum*) the median intergenic region is around 2kb or less [93]. The versions of the different genomes are the following:

- *P. falciparum*: 3D7, Jun 18, 2015

- *P. berghei*: ANKA, Jan 09, 2017

- *T. gondii*: TGME49, 2015-03-22

- *S. cerevisiae*: R64

- Human: GRCh37

- *Drosophila*: BDGP6

- *C. elegans*: WBcel235

- *A. thaliana*: TAIR10

- *T. thermophila*: JCVI-TTA1-2.2

- *D. discoideum*: dicty_2.7

## In vivo experimental validation in *P. falciparum*

**Cloning of DNA constructs.** All PCR amplifications were done with high-fidelity polymerase PfuUltra II Fusion HS DNA Polymerase (Agilent Technologies) following the recommended protocols, except we lowered the elongation temperature to 62˚C. All cloning reactions used the In-Fusion® HD Cloning Kit (Takara Bio USA, Inc.) and followed the manufacturer's protocol. All PCR and digestions were purified using PCR Clean Up Kit (Macherey-Nagel) and followed the manufacturer's protocol, except we performed 4-6 washes using 700µL buffer NT3. All cloning and plasmid amplifications were done in Escherichia coli, XL10-Gold Ultracompetent Cells (Stratagene). All minipreps and maxipreps were performed using NucleoSpin Plasmid, Mini kit for plasmid DNA (Macherey-Nagel), and NucleoBond Xtra Maxi Plus kit for transfection-grade plasmid DNA (Macherey-Nagel), respectively. Sanger sequencing confirmed the absence of undesired mutations in the homology regions, the guide sequence, and the recombinant region.

The plasmids pBLD587_3rep_PfGFP and pBLD587_1rep_PfGFP were constructed in multiple cloning steps from a pBLD587_HAtag-GFP backbone containing a Pfs47 homology region [78]. The 3' UTR from *P. falciparum* HRPII was amplified from a PCC1 plasmid [94] and cloned using primers 1 and 2 (see Table 2) and restriction sites SpeI and HindIII. The PfGFP was designed by codon optimization of the GFP using codon usage tables for *P. falciparum* 3D7 from Codon Usage Database (http://www.kazusa.or.jp/codon), minimizing GU wobble pairings [95], and adding the 15-bp homology necessary for InFusion cloning. PfGFP was ordered as a DNA gBlocks® gene fragment from Integrated DNA Technologies and cloned using restriction sites HindIII and BamHI. Next, the core promoter and 356 bp DNA fragment that includes the identified ATA-enriched region of PF3D7_0913900 was amplified from *P. falciparum* 3D7 genomic DNA and cloned using primers 3 and 4 and restriction sites XhoI and BamHI resulting in the plasmid pBLD587start. To generate the pBLD587_1rep_PfGFP plasmid, we amplified 712 bp PF3D7_0805300 5' intergenic region from *P. falciparum* 3D7 gDNA and cloned it using primers 5 and 6 and restriction sites NotI and XhoI into the plasmid pBLD587start. The final length of the 5' regulating sequence of its recombinant construct was 1246 bp for a GC content

**Table 2. Primers used for cloning.** Nucleotides in lowercase show the overhangs introduced into oligonucleotides that are necessary to use InFusion cloning.

| 1 | tttatagtacactagTCTTATATATAATGAG |
|---|---|
| 2 | atattatgtaaagcttAGCTTATTTAATAATAG |
| 3 | tagaaaccatggatccTATCCATTATGTATAAAAC |
| 4 | cgttatgttactcgagTGAAAATTATCAGGAAATAAAAC |
| 5 | ataattttcactcgagCATATATGTGTATAAATAAAAACAC |
| 6 | accgcggtggcggccgcaAAATAACATAAATATAAATG |
| 7 | cataacgtaaccggtcttaaGTTTTCTTCGTTACATG |
| 8 | accgcggtggcggccgcTGAAAATTATCAGGAAATAAAAC |
| 9 | ataattttcactcgagTTTTCTTCGTTACATG |
| 10 | acgaagaaaacttaagTGAAAATTATCAGGAAATAAAAC |

of 15,97%. To generate the pBLD587_3rep_PfGFP plasmid, the second 356 bp ATA-enriched region from PF3D7_0913900 was amplified and cloned using primers 7 and 8 and restrictions sites AflII and NotI into pBLD587start plasmid. Next, the third 356 bp ATA-enriched region was amplified from previous 3' part of PF3D7_0913900 5'UTR amplicon and cloned using primers 9 and 10 and restriction sites XhoI and AflII. The final length of the 5' regulating sequence of its recombinant construct was 1250 bp for a GC content of 12.56%.

**Parasite culture and transfection.** *P. falciparum* 3D7 strain (MR4, ATCC) was cultivated in complete RPMI containing 5% human serum and 0.5% Albumax II (Thermo Fisher Scientific) and A-type human blood at 37˚C, 5% $CO_2$, 5% $O_2$ under agitation. Synchronous parasites were obtained by treating infected red blood cells with 9 volumes of 5% sorbitol for 10 min at 37˚C. After one cycle, rings were used for transfections, using 60 $\mu$g of each plasmid (pBLD587_3rep_PfGFP and pBLD587_1rep_PfGFP) plus 60 $\mu$g of pfs547 (kindly provided by Christiaan van Ooij), containing Cas9 and the Pfs47 guide RNA. Ring transfection was performed as published [96] with one pulse at 310 V, 950 $\mu$F in a GenePulser Xcell (Bio-Rad) in 0.2 cm cuvettes (Bio-Rad). After electroporation, parasites were cultivated for one day without drug selection, followed by 5 days in which media containing 2.5 nM WR99210 was changed daily, and then every 2 days until drug-resistant parasites appeared in the cultures. Once integration was confirmed by PCR using primers checkPfs47F (CATTCCTAACACATTATGTG TATAACATTTTATGC) and checkPfs47R (CATATGCTAACATACATGTAAAAAATTAC AATCAG), parasites were cultivated without drug pressure and cloned to obtain episome-free parasites.

**qPCR analysis.** For qPCR analysis, late-stage parasites were purified by gelatin flotation (Goodyer ID et al. 1994 Ann Trop Med Parasitol) and left to reinvade for 6 hours, after which a sorbitol treatment was applied to eliminate parasites that did not reinvade, allowing only young rings to continue the cycle. After one cycle, rings at 14 hours post-invasion, as confirmed by Giemsa staining, were collected by 0.15% saponin lysis of RBCs, RNA was extracted by Trizol (Invitrogen), quantified by NanoDrop (Ozyme) and 1 $\mu$g of total RNA was reverse-transcribed using Superscript IV (Invitrogen / Life technologies). Quantitative PCR was done in a LightCycler480 (Roche—Plateforme qPHD UM2 / Montpellier GenomiX) using Power SyBr Green PCR Master Mix (Thermo Scientific) and GFP primers (GFP_F1 TACCCAATG TAATACCGGCGG, GFP_R1 GGTGACGGACCAGTTTTGTTG, GFP_F4 ACAAGAGTGT CTCCCTCGAAC, GFP_R4 CCTGTGCCATGGCCTACTTTA), and as normalizers, seryl-tRNA synthetase (PF3D7_0717700), fructose-biphosphate aldolase (PF3D7_1444800) [97]. Relative copy numbers were obtained from standard curves of genomic DNA of integrated parasites. Two clones were used for each transgenic parasite line.

## Supporting information

**S1 Fig. Accuracy of DExTER models trained using only di-nucleotides on whole region.**
(PDF)

**S2 Fig. Accuracy of models trained using TF motifs, with or without addition of DExTER variables.**
(PDF)

**S3 Fig. Proportion of isolated occurrences of identified k-mers in regions.**
(PDF)

**S4 Fig. Length of repetitive blocks.**
(PDF)

**S5 Fig. Correlations between expression profiles of conditions.**
(PDF)

**S6 Fig. Variable importance in the different models.**
(PDF)

**S7 Fig. Prediction of *P. falciparum* gene expression during erythrocytic cycle, with convolutional neural networks.**
(PDF)

**S8 Fig. Gene Ontology enrichment analysis of 5 important variables of *P. falciparum*.**
(PDF)

**S9 Fig. DExTER accuracy for predicting H2AZ, H3K9ac and H3K4me3 histone marks.**
(PDF)

**S10 Fig. Correlations between histone mark signal and k-mer frequency of the most important variables identified for the different histone marks and time points.**
(PDF)

**S11 Fig. Prediction of sense and antisense transcript levels.**
(PDF)

**S12 Fig. Control of transcriptional activity of one housekeeping gene in the two genetically modified parasites.**
(PDF)

**S1 Appendix. best.features.zip.** This zip file is an archive that contains the most important cLREs identified in the different experiments for *P. falciparum*, *P. berghei*, *T. gondii*, *S. cerevisiae*, human, *A. thaliana*, *Drosophila*, *C. elegans*, *D. discoideum* and *T. thermophila*. Each file is a table (in text mode) where genes are in rows, and cLREs in columns. These tables can be used to rank the genes with respect to a chosen cLRE in order to identify the genes for which the k-mer frequency is the highest/lowest in the associated region.
(ZIP)

## Acknowledgments

We thank PlasmoDB for the invaluable malaria-database support, and Plateforme qPHD UM2 / Montpellier GenomiX for the Roche thermocyclers.

## Author Contributions

**Conceptualization:** Christophe Menichelli, Vincent Guitard, Sophie Lèbre, Jose-Juan Lopez-Rubio, Charles-Henri Lecellier, Laurent Bréhélin.

**Funding acquisition:** Jose-Juan Lopez-Rubio, Charles-Henri Lecellier, Laurent Bréhélin.

**Methodology:** Christophe Menichelli, Sophie Lèbre, Charles-Henri Lecellier, Laurent Bréhélin.

**Software:** Christophe Menichelli.

**Supervision:** Jose-Juan Lopez-Rubio, Laurent Bréhélin.

**Validation:** Christophe Menichelli, Vincent Guitard, Rafael M. Martins, Jose-Juan Lopez-Rubio, Charles-Henri Lecellier, Laurent Bréhélin.

**Writing – original draft:** Christophe Menichelli, Vincent Guitard, Jose-Juan Lopez-Rubio, Charles-Henri Lecellier, Laurent Bréhélin.

**Writing – review & editing:** Rafael M. Martins, Sophie Lèbre.

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
