## [Decision Letter · Decision Letter 0]

21 Nov 2020

Dear Dr. Brehelin,

Thank you very much for submitting your manuscript "Identification of long regulatory elements in the genome of Plasmodium falciparum and other eukaryotes" for consideration at PLOS Computational Biology.

As with all papers reviewed by the journal, your manuscript was reviewed by members of the editorial board and by several independent reviewers. In light of the reviews (below this email), we would like to invite the resubmission of a significantly-revised version that takes into account the reviewers' comments.

We cannot make any decision about publication until we have seen the revised manuscript and your response to the reviewers' comments. Your revised manuscript is also likely to be sent to reviewers for further evaluation.

Sincerely,

Ilya Ioshikhes

Associate Editor

PLOS Computational Biology

Erik van Nimwegen

Deputy Editor

PLOS Computational Biology

Reviewer's Responses to Questions

**Comments to the Authors:**

Reviewer #1: Dear editor,

The author Menichelli et al has reported a method to identify long regulatory elements in Plasmodium falciparum and some other eukaryotes. The paper is of interest to the research community of apicomplexan research. I recommend the authors compile their findings, and give concrete numbers of LRE identified and make them easily accessible for other. Additionally, however, I have the following remarks and questions:

1.DexTER, the acronym for the method, was not defined until much later in the manuscript. A brief description of one or two sentences, of the method should be mentioned in the abstract.

2.“LERs appear to determine a very large part of gene expression variation”, page 2. This is not precise language for reporting science. Can the authors define “a very large part”

3.“For most specie, we show that these LRE are predictive of expression with an accuracy in between 50% and 60%, for P. falciparum, the accuracy even exceeds 70%”, page 5, these numbers 50%, 60% 70%, do not have a context. Please provide bench marking and rationale for the accuracy, because these number will determine the quality of the research

4.Related to 3, also page 7&8, please provide a benchmark comparison, how much can be recovered, which category, of LERs with at least one other species, e.g. human.

5.“We took 4kb centered either around the gene start, or around AUG ..”, page 6, please provide rationale for choosing 4000 bp.

6.Is there a supplemental file with all the LERs characterized? So that other researchers can perform independent validations. And the author’s work will be more accessible for the community.

7.Fig 5, why is P. falciparum’s result so different than that of P. berghei?

Reviewer #2: Summary comments:

Overall this is an interesting manuscript that explores the association of long repeat elements (LRE) with gene expression in a number of organisms. The main focus of the work is developing a systematic method to discover LREs adjacent to known or predicted genes and then to assess the association of those sequences with gene expression profiles. For most organisms, the association fluctuates around 60%, but in the case of Plasmodium falciparum is higher and this manuscript focuses on the implications for P. falciparum gene expression. In P. falciparum, whose genome is known to be strongly depleted of transcription factors, LREs appear to determine a very large part of gene expression variation, and their analyses show that they are involved in both transcriptomic and post-transcriptomic regulation mechanisms.

This work is novel and does provide an interesting set of correlative data for the potential role of long repeat elements in the regulation of P. falciparum gene expression. This is quite intriguing because there remain major questions as to the mechanism of gene regulation. There is a very structured program of gene regulation throughout the life cycle and a paucity of predicted transcriptional regulatory proteins. While the work presented here is of potential interest, a major question as to the relevance of these findings remains. The Plasmodium falciparum genome has an unbalanced base composition with 82% A+T on average across the genome. Most of the G+C pairs occur in genic regions and parts of the non-genic genome reach nearly 100% A+T with long stretches of homopolymer and low complexity (AT repeats) sequence surround genes. The comparisons to other organisms did not include organisms with similar high A+T content. This could influence the results. Of particular concern is the identified key LREs are TATAT, TTT, TATATA and AA all seem to show strong correlations.

On the positive side, the authors do show strong associations with particular sequences with certain biological features of gene expression in the parasite and the computational model may be teasing out relationships not previously recognized.

The authors extend their analysis to a functional test of the LREs by constructing two chimeric upstream regions, one with high ATA content and one with low ATA content attached to a GFP reporter and then inserting these in the Plasmodium pfs47 gene locus and measuring GFP expression. They observed a 10-fold difference in expression and while potentially interesting probably will require some additional characterization. Pfs47 is sexual stage specifically expressed in its native configuration so the CRISPR/Cas9 inserted construct has in some way changed the native expression profile – perhaps through the disruption of chromatin structure. Before definitive conclusions can be drawn, including additional work to determine if this difference in expression is context or sequence dependent. It would be particularly interesting of insertion of a specific LRE disrupted chromatin structure.

Overall, this work is of potential interest and may shed light on the mechanisms associated with the control of gene expression. The work is innovative and novel and has opened some new questions that merit further experimental exploration.

Reviewer #3: In the manuscript entitled “Identification of long regulatory elements in the genome of Plasmodium falciparum and other eukaryotes” by Menichelli et al., authors have used a computational approach to identify the long regulatory elements (LREs) in the genomes of various eukaryotic systems. They have also used various conditions to train the program to eventually use it to identify the LREs and predict the gene expression from unknown sequences. Authors have used expression data as a readout for the direct association of the long regulatory elements with the regulation of the associated genes. Finally, they have validated the association between the LREs and gene expression in Plasmodium falciparum using two LREs of varying length in in vitro conditions. In this study, investigators have addressed an important long-standing question in the field and identified novel LREs and their association with gene expression. While the analysis is rigorous and potentially very interesting the manuscript needs clarity in presentation and explanation. Here are some of the concerns outlined.

1. One of the major concerns is that authors have used the association between LREs and gene expression as causation (i.e. regulation) interchangeably throughout the manuscript. Though causation is possible, the data provided in this manuscript is not sufficient to establish the causal relationship.

2. The correlation (which I believe is Pearson correlation) between LREs and expression is generally in the range of 0.2 to 0.5 in Figure 4. These are mostly weak or moderate associations.

3. Most of the housekeeping genes in higher eukaryotic systems (human and mouse) have high CpG content at promoters. Thus sequence itself is governing the gene expression in this case. How do the authors explain the sinusoidal pattern of gene expression in Plasmodium with LREs?

4. I was wondering if DExTER has also identified CpG elements in the human/mouse genome? Does it correlate with gene expression?

5. The design for in vitro experimental validation is not clear. In this experiment, the authors have used two different sequences and compared the GFP expression. As these are two different sequences they cannot be directly compared. Authors should take varying repetitions of the ATA sequences (eg. 3, 2, and 1) to see the effect of this LTR on expression. Moreover, if they want to demonstrate this association for causation, they need to mutate the LRE of the same length.

6. It would be important that authors provide information pertinent to the important LREs identified in a tabulated form. LRE-sequence/associated with number of genes/LRE count/gene ontology/significance value.

7. Which version of the Plasmodium genome is used for the analysis?

8. Authors have also looked at conservation of the identified LREs along evolution. Why do authors believe that there will be conservation in LREs when genomes are so different? Please include an explanation.

9. The text in figure 1 is very hard to read due to the very small font size.

10. Axis label is missing in figure 2. Also, tissue type/stages of development are not explained properly in the figure or the text.

11. Figure 4: The ‘most important variables’ indicated on the graph are hard to read due to the poor contrast and small font size.

12. In general, the font sizes of the axis labels are too small for many supplementary figures. Figure caption is missing for figure S3. Axis label is missing in Figure 7a, S5 and S9.

13. Authors should specify moderate/ average/higher gene expression correlations.

14. The assumption that LREs are involved in post-transcriptional regulation is not supported in this study. Plasmodium also produces a lot of anti-sense RNAs and these LREs might have a role in transcriptional regulation of anti-sense RNAs.

**Have all data underlying the figures and results presented in the manuscript been provided?**

Reviewer #1: None

Reviewer #2: Yes

Reviewer #3: Yes

PLOS authors have the option to publish the peer review history of their article (what does this mean?). If published, this will include your full peer review and any attached files.

Reviewer #1: No

Reviewer #2: No

Reviewer #3: No
---

## [Decision Letter · Decision Letter 1]

18 Feb 2021

Dear Dr. Brehelin,

Thank you very much for submitting your manuscript "Identification of long regulatory elements in the genome of Plasmodium falciparum and other eukaryotes" for consideration at PLOS Computational Biology.

As with all papers reviewed by the journal, your manuscript was reviewed by members of the editorial board and by several independent reviewers. In light of the reviews (below this email), we would like to invite the resubmission of a significantly-revised version that takes into account the comments of reviewer 2.

We cannot make any decision about publication until we have seen the revised manuscript and your response to the reviewers' comments. Your revised manuscript is also likely to be sent to reviewers for further evaluation.

Sincerely,

Ilya Ioshikhes

Deputy Editor

PLOS Computational Biology

Erik van Nimwegen

Deputy Editor

PLOS Computational Biology

Reviewer's Responses to Questions

**Comments to the Authors:**

Reviewer #1: Most of my questions have been answered or addressed.

Reviewer #2: The authors have responded with new data and a modified text to the comments by the referees. The addition of the analysis of other high A+T content organisms provides additional new data and strengthens the manuscript. There remains one issue that was not adequately addressed. The functional validation remains a weak point in the manuscript. The authors have not addressed issues of genomic context - in particular, the authors have used a gene that is normally only expressed in the mosquito stages and yet they measure the transcription in the asexual stages. There are no controls presented to demonstrate that transcriptional levels of other genes have similar transcriptional levels in the two genetically modified parasites. There is not confirmation that the two lines are assayed in parasites at the same stage of the life cycle. Without additional experiments including key control experiments, the data presented cannot be used as proof of validation. This is the beginning of a validation process, but far from the acceptable standard.

Reviewer #3: In th manuscript entitled “Identification of long regulatory elements in the genome of Plasmodium falciparum and other eukaryotes” by Menichelli et al., Authors have satisfactorily answered all my concerns.

**Have all data underlying the figures and results presented in the manuscript been provided?**

Reviewer #1: None

Reviewer #2: Yes

Reviewer #3: None

PLOS authors have the option to publish the peer review history of their article (what does this mean?). If published, this will include your full peer review and any attached files.

Reviewer #1: No

Reviewer #2: No

Reviewer #3: No
---

## [Decision Letter · Decision Letter 2]

14 Mar 2021

Dear Dr. Brehelin,

Thank you very much for submitting your manuscript "Identification of long regulatory elements in the genome of Plasmodium falciparum and other eukaryotes" for consideration at PLOS Computational Biology. As with all papers reviewed by the journal, your manuscript was reviewed by members of the editorial board and by the Reviewer #2 who recommended in the confident comments to the editor several changes that must be implemented in order for the manuscript be processed further. The reviewer appreciated the attention to an important topic. Based on the confidential comments, we are likely to accept this manuscript for publication, providing that you modify the manuscript according to the reviewers recommendations.

Please have the additional data presented only for the review be included in supplemental material.  We are also asking you to qualify the conclusions you draw from the validation experiments, citing the many caveats to this work, which were previously mentioned.  Specifically lines 539-541 are a broad overstatement.  Lines 478-479 should be deleted - at best these results indicate that there should be further investigation of these LREs for a potential role in transcriptional regulation.  We recommend to delete the last line of the Abstract.

Sincerely,

Ilya Ioshikhes

Deputy Editor

PLOS Computational Biology

Erik van Nimwegen

Deputy Editor

PLOS Computational Biology

[LINK]

Reviewer's Responses to Questions

**Comments to the Authors:**

Reviewer #2: none

**Have all data underlying the figures and results presented in the manuscript been provided?**

Reviewer #2: Yes

PLOS authors have the option to publish the peer review history of their article (what does this mean?). If published, this will include your full peer review and any attached files.

Reviewer #2: No

Figure Files:

Data Requirements:

Reproducibility:

References:

---

## [Decision Letter · Decision Letter 3]

24 Mar 2021

Dear Dr. Brehelin,

We are pleased to inform you that your manuscript 'Identification of long regulatory elements in the genome of Plasmodium falciparum and other eukaryotes' has been provisionally accepted for publication in PLOS Computational Biology.

Best regards,

Ilya Ioshikhes

Deputy Editor

PLOS Computational Biology

Erik van Nimwegen

Deputy Editor

PLOS Computational Biology

Reviewer's Responses to Questions

**Comments to the Authors:**

Reviewer #2: none

**Have all data underlying the figures and results presented in the manuscript been provided?**

Reviewer #2: Yes

PLOS authors have the option to publish the peer review history of their article (what does this mean?). If published, this will include your full peer review and any attached files.

Reviewer #2: No

---

## [Editor Report · Acceptance letter]

13 Apr 2021

PCOMPBIOL-D-20-01622R3 

Identification of long regulatory elements in the genome of *Plasmodium falciparum* and other eukaryotes

Dear Dr Bréhélin,

I am pleased to inform you that your manuscript has been formally accepted for publication in PLOS Computational Biology. Your manuscript is now with our production department and you will be notified of the publication date in due course.

With kind regards,

Katalin Szabo
